# CheMixHub: Datasets and Benchmarks for Chemical Mixture Property Prediction

**Ella Miray Rajaonson**[1,2]    **Mahyar Rajabi Kochi**[1]    **Luis Martin Mejía Mendoza**[3]

**Seyed Mohamad Moosavi**[1,2]    **Benjamin Sanchez-Lengeling**[1,2]

`ben.sanchez@utoronto.ca`

[1] University of Toronto, Canada
[2] Vector Institute for Artificial Intelligence, Canada
[3] Clean Energy Innovation Research Center, National Research Council, Canada

## Abstract

Developing improved predictive models for multi-molecular systems is crucial, as nearly every chemical product used results from a mixture of chemicals. While being a vital part of the industry pipeline, the chemical mixture space remains relatively unexplored by the Machine Learning (ML) community. In this paper, we introduce CheMixHub, a holistic benchmark for molecular mixtures spanning a corpus of 11 chemical mixtures property prediction tasks. With applications ranging from drug delivery formulations to battery electrolytes, CheMixHub currently totals approximately 500k data points gathered and curated from 7 publicly available datasets. We devise various data splitting techniques to assess context-specific generalization and model robustness, providing a foundation for the development of predictive models for chemical mixture properties. Furthermore, we map out the modelling space of deep learning models for chemical mixtures, establishing initial benchmarks for the community. This dataset has the potential to accelerate chemical mixture development, encompassing reformulation, optimization, and discovery. The dataset and code for the benchmarks can be found at: https://github.com/chemcognition-lab/chemixhub

## 1 Introduction

Mixtures of molecules are integral to our daily experiences: from the perfumes we smell [59] to the remedies we take [3, 75]. Understanding the interactions and behaviors of molecular mixtures is therefore essential for advancements in biochemistry [17], drug discovery [3] and environmental science [24]. Mixtures are particularly appealing because they offer greater flexibility in tailoring properties that specific application needs. By adjusting composition, substituting components, or introducing new ones, it is possible to fine-tune characteristics such as viscosity [6], volatility [19], stability [37], and conductivity [32]—features that are highly task-dependent. However, discovering new chemical mixtures, optimizing or reformulating them requires thorough characterization, which is time- and resource-intensive due to the exponentially growing number of candidate combinations. The mixture search space is vastly larger than that of single-component systems, making exhaustive experimental exploration impractical.

While ML has emerged as a powerful tool for accelerating the characterization and discovery of new materials [50], it faces unique challenges in the context of mixtures. On one hand, mixtures properties are highly correlated with strength of intermolecular interactions that cannot be inferred directly from the behavior of individual components. This causes quantitative structure–property relationship (QSPR) modeling remain underexplored for multi-component systems compared to the

substantial progress achieved for single-component systems [47]. On the other hand, the scarcity of publicly available datasets further limits progress in this area. While mono-molecular systems have benefited from well-established, community-driven datasets and benchmarks—such as MoleculeNet [69] and the Therapeutics Data Commons [25]—no centralized or standardized database currently exists for multi-component molecular systems.

In this paper, we introduce CHEMIXHUB, the first comprehensive benchmark of tasks for property prediction in chemical mixtures (see Figure 1). Covering 11 tasks across diverse chemical domains, CHEMIXHUB aims to enable systematic exploration of critical research questions, including:

1. What modeling strategies, particularly those exploiting permutation invariance and hierarchical structure, are most effective for mixtures?

2. Can a general-purpose representation generalize effectively across multiple mixture tasks?

3. To what extent does incorporating physics-based constraints improve model performance, particularly in varying experimental contexts like temperature?

Our key contributions, designed to facilitate the investigation of these questions, include:

- **Dataset curation**: Consolidating and standardizing 11 tasks from 7 datasets, reflecting the diversity and state of the chemical mixtures space.

- **New tasks**: Introducing two new tasks from a new dataset, curated from the IlThermo database (116,896 data points) and providing code to easily extend the curation to other target properties.

- **Generalization splits**: Implementing four distinct data splitting methodologies (random, unseen chemical component, varied mixture size/composition, and out-of-distribution context) to enable robust assessment of model generalization capabilities under various realistic scenarios.

- **Establishing baselines**: Benchmarking representative ML models to set initial performance levels and provide a comparative framework for future development.

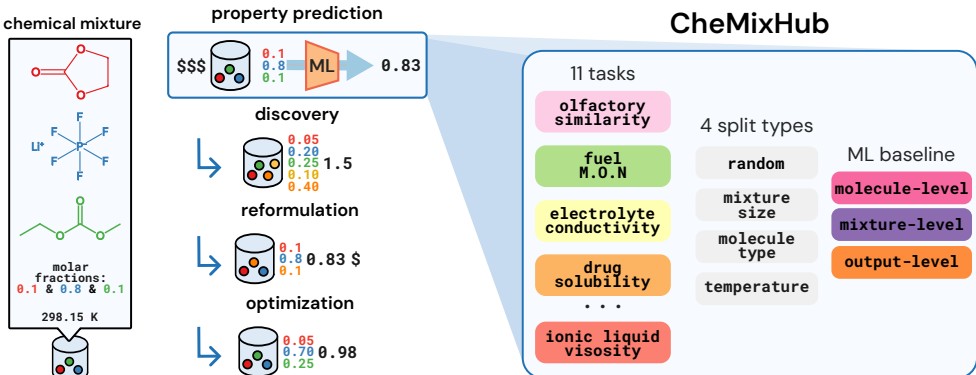

Figure 1: **CheMixHub: A benchmark for chemical mixture property prediction.**. (Left) Illustrates a sample mixture input, including components and conditions. (Center) Highlights potential applications enabled by CheMixHub, such as reformulation, optimization, and discovery through property prediction. (Right) Summarizes CheMixHub's structure: 11 tasks, 4 data split types, and a multi-level modeling baselines for comprehensive evaluation and development.

## 2 Related works

**Chemical mixture properties datasets**  Various open-access and commercial sources offer experimental and computational mixture data, though these predominantly cover binary and ternary systems, with complex multi-component mixtures underrepresented (see Figure 2) [60, 45, 74, 35]. For instance, the open-source NIST ILThermo database [27] provides temperature-dependent transport and thermophysical properties for ionic liquid mixtures. Commercial platforms like DETHERM [68] and the Dortmund Data Bank [41] offer thermophysical data but are not publicly accessible. Despite these resources, substantial mixture data remains scattered throughout the literature, highlighting the need for unified datasets to enable robust mixture behavior modeling [81, 7, 14] and drive progress.

**Deep learning on sets** Predicting mixture properties from a set of components requires models that respect key inductive biases, notably permutation invariance [70] [23]. Many permutation-invariant set functions follow a common blueprint: a series of permutation-equivariant operations (e.g., element-wise MLPs or self-attention layers) followed by a permutation-invariant aggregation (e.g., summation or pooling) [8]. The pioneering *DeepSets* architecture [73, 42] exemplifies this, using an element-wise MLP and sum aggregation, followed by further non-linear processing ( *sum-decomposition* [52, 57, 48, 65, 39, 84]). SetTransformer [34, 13] uses self-attention to model pairwise interactions with attention-based aggregation. Janossy pooling [40] offers another approach, explicitly modeling invariance and capturing higher-order interactions by averaging outputs over multiple input permutations [70, 79].

**Learning on chemical mixtures** ML is increasingly applied to chemical mixtures, primarily for property prediction, with emerging work on optimization and discovery [83, 54]. Common strategies involve aggregating molecular-level chemo-informatic features, graph neural networks (GNNs) embeddings [55, 76, 59, 81, 6] or large pre-trained chemical language models (CLMs) [56, 46]—using a *DeepSets*-like architecture. Alternatively, mixture representations are formed by weighted combinations of individual component descriptors [55, 56, 46], or learned by tree-based models from these descriptors [3]. A separate line of research explores attention-based architectures [76, 59]. Some work explicitly models pairwise interaction terms between mixture components, offering a more physically grounded and expressive representation [81]. These mixture embeddings are then fed to predictive models like neural networks [6], gradient boosting machines [3], or Gaussian processes [54].

## 3 Dataset

### 3.1 Corpus overview

We curated 11 regression tasks from 7 published datasets across the chemical mixture literature, which are summarized in Table 1. The tasks were selected based on application domain and prior use as baselines in ML studies. To ensure their accessibility and standardization for ML applications, we provide a Croissant [2] file detailing their metadata, structure, and semantics based on schema.org. The dataset licenses are listed in Appendix A.1. Additional statistics on the molecules found in each task are provided in Appendix A.2.

Table 1: **CheMixHub tasks summary**. *T* indicates temperature dependency. *Mole Fractions* indicates availability of mole fraction information. *Arrhenius relationship* indicates if the target property can be modeled using the Arrhenius equation. *Exp.* indicates if the data was obtained from wet-lab experiments or simulations.

| Dataset | Tasks | Units | Datapoints | Max # Components | # Unique Mixtures | # Unique Molecules | Mixture Context | Mole Fractions | Arrhenius Relationship | Exp. |
|---|---|---|---|---|---|---|---|---|---|---|
| Miscible Solvents | $\rho$ | g/m$^3$ | 30,142 | 5 | 19,238 | 81 | — | ✓ | ✗ | ✗ |
| | $\Delta H_{\mathrm{mix}}$ | kJ/mol | 30,142 | 5 | 19,238 | 81 | — | ✓ | ✗ | ✗ |
| | $\Delta H_{\mathrm{vap}}$ | kcal/mol | 30,142 | 5 | 19,238 | 81 | — | ✓ | ✗ | ✗ |
| IlThermo (Ionic Liquids) | $\ln(\kappa)$ | S/m | 40,904 | 3 | 14,438 | 479 | T | ✓ | ✓ | ✓ |
| | $\ln(\eta)$ | Pa·s | 75,992 | 3 | 15,878 | 699 | T | ✓ | ✓ | ✓ |
| NIST Viscosity (Liquid mixtures) | $\ln(\eta_{\mathrm{NIST-full}})$ | cP | 239,201 | 2 | 84,133 | 1648 | T | ✓ | ✓ | ✓ |
| | $\ln(\eta_{\mathrm{NIST}})$ | cP | 34,374 | 2 | 4566 | 1397 | T | ✓ | ✓ | ✓ |
| Drug solubility | $\ln(S)$ | g/100g | 27,166 | 3 | 3259 | 169 | T | ✓ | ✗ | ✓ |
| Solid Polymer Electrolytes | $\ln(\kappa)$ | S/m | 11,350 | 5 | 1749 | 402 | T | ✓ | ✓ | ✓ |
| Olfactory mixtures | Perceptual similarity | — | 865 | 43 | 743 | 201 | — | ✗ | ✗ | ✓ |
| Fuel mixtures | MON | — | 684 | 121 | 352 | 419 | — | ✓ | ✗ | ✓ |

### 3.2 Datasets & Tasks

**Miscible Solvents (3 tasks)** Homogeneous solutions are important in a variety of material science applications such as battery electrolytes, chemical reactivity, and consumer packaged goods. The Miscible Solvents dataset provides a set of three tasks centered around miscible solvent properties, originally generated by Chew et al. using molecular dynamics (MD) simulations for 19,238 unique mixtures [14].

- **Density ($\rho$):** $\rho$ measures how tightly packed the molecules are in a mixture. In industrial applications, density is important as it dictates the final weight and polarity of the product.

- **Heat of vaporization ($\Delta H_{\mathrm{vap}}$):** $\Delta H_{\mathrm{vap}}$ is the amount of heat needed to convert some fraction of liquid into vapor. While experimentally measuring $\Delta H_{\mathrm{vap}}$ for mixtures is challenging, it

effectively measures the cohesion energy of a liquid and has been previously observed to correlate with temperature-dependent viscosity [15].

- **Enthalpy of mixing ($\Delta H_{\text{mix}}$):** $\Delta H_{\text{mix}}$ is a fundamental thermodynamic property of liquid mixtures that measures the energy released or absorbed upon the mixing of pure components into a single phase in equilibrium. It is important for process design that dictates properties, such as solubility and phase stability.

**IlThermo (2 tasks)** Ionic liquids (ILs) are salts composed of organic cations and organic or inorganic anions that remain liquid at temperatures below 100 °C [1, 62, 80]. ILThermo is a web-based database that provides extensive information on over 50 chemical and physical properties of pure ILs, as well as their binary and ternary mixtures with various solvents [28]. For the scope of this paper, we selected two property prediction tasks from IlThermo; however, we have open-sourced our curation code to facilitate the addition of further tasks in the future. Details of the curation process are provided in Section A.3, and the selected tasks are summarized below:

- **Ionic conductivity ($\kappa$):** Higher $\kappa$ makes ILs attractive for use as electrolytes in energy storage and other electrochemical applications [38]. However, ionic conductivity of ionic liquid mixtures is a complex phenomenon and is influenced by multiple factors such as size and charge on the ions, polarity and dielectric strength of the solvent, viscosity, hydrogen bonding strength, ion association, etc [58]. To facilitate data-driven approaches, IlThermo dataset includes 40,904 $\kappa$ data points curated from literature, covering 14,438 unique mixtures composed of 479 distinct molecules.

- **Viscosity ($\eta$):** Modeling the viscosity of ILs is particularly challenging, as their viscosity can be orders of magnitude higher than those of conventional solvents [44]. This complexity arises from the coexistence of multiple interaction types—ionic, dispersion, dipole-dipole, and induced dipole interactions—that are more pronounced compared to typical organic solvents. The ILThermo dataset provides 75,992 viscosity ($\eta$) data points curated from the literature, encompassing 15,878 unique mixtures formed from 699 distinct molecules.

**NIST viscosity (2 task)** Dynamic viscosity is a key design objective for modern process engineering and products. However, modeling the viscosity of liquid mixtures presents significant challenges due to the complex molecular interactions and the potential for nonmonotonic behavior [51]. NIST Thermodynamics Research Center (TRC) data archival system provides one of the most comprehensive datasets containing 239,201 dynamic viscosity datapoints of binary liquid mixtures [27]. Bilodeau et al. proposed a smaller version of the dataset by applying two key preprocessing steps: (1) removing data entries with SMILES strings containing multiple, non-covalently bonded fragments, and (2) excluding entries where either molecule was predicted to be a gas or solid in its pure form. These steps reduced the dataset to 34,374 data points[6]. We include both version of the dataset and refer to each as NIST-full and NIST, respectively.

**Drug solubility (1 task)** The drug solubility in mixture of solvents is a critical factor that influences various stages of the pharmaceutical development pipeline, from drug discovery, drug analysis to formulation design. It allows greater flexibility through adjusting solvent combinations and ratios enabling solubility to be tailored to meet specific needs and to co-dissolve other necessary materials. The dataset was originally curated by Bao et al. from literature and includes 27,166 data points [3]

**Solid polymer electrolytes ionic conductivity (1 task)** Solid polymer electrolytes (SPEs), proposed as potential replacements for conventional liquid organic electrolytes in batteries, have been engineered to offer improved electrochemical stability and reduced flammability. However, their practical use is limited by inherently low ionic conductivity. To support research on this issue, Bradford et al. compiled a dataset from the literature comprising 11,350 ionic conductivity measurements across more than 1,700 unique electrolyte formulations. Each formulation is uniquely defined by the polymer, salt, salt concentration, polymer molecular weight, and any additives present [7].

**Fuel mixture Motor Octane Number (1 task)** Kuzhagaliyeva et al. compiled a database containing 684 data points for 352 unique single hydrocarbons and mixtures, reporting experimentally measured motor octane numbers (MON) from various literature sources. The MON is a combustion-related property commonly used to assess a fuel's resistance to knocking. The dataset is categorized into three subpopulations: pure components, blends with 10 or fewer components (mostly surrogates), and complex fuels containing more than 10 components [31].

**Olfactory mixture perceptual similarity (1 task)** Predicting the perceptual similarity of olfactory mixture contributes to olfaction digitization efforts [33] and also enables mixture reformulation. The dataset was originally compiled from previous publications [67, 53, 9] by Tom et al. [59]. Data for each of these publications was obtained from pyrfume [11] and consists of 865 pairwise mixture comparisons. Each pair is assigned a continuous perceptual similarity score ranging from 0 (completely similar) to 1 (completely different). This final score represents an average of similarity ratings obtained from human participants across different experimental paradigms.

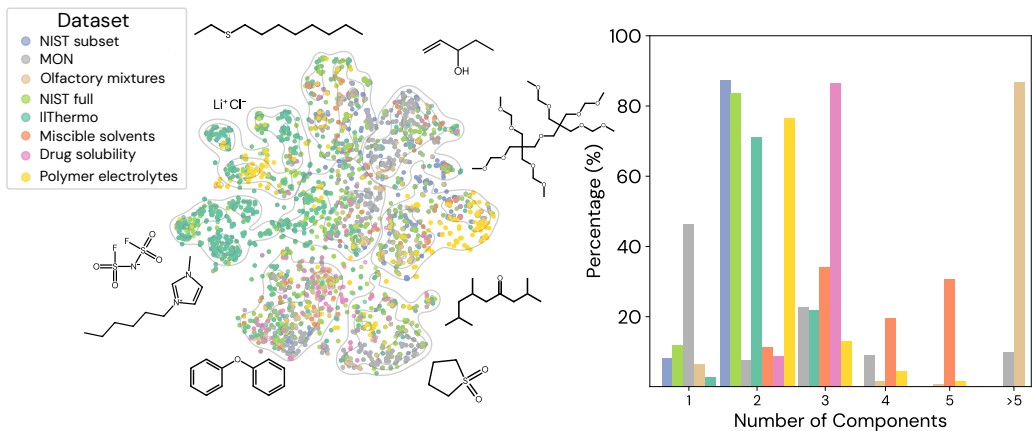

Figure 2: **Diversity of Chemical Structures and Mixture Compositions in CheMixHub**. (Left) t-SNE visualization of the molecular structural diversity, with points colored by their source dataset. (Right) Histogram showing the percentage of mixtures based on their number of components.

### 3.3 Curation pipeline

The following points highlight data set design choices we made:

- **Handling of chemical species diversity** The diversity of the chemical-mixture space extends beyond fields of applications to the fundamental level of chemical representation itself. Indeed, different chemical moieties for which representations may not have been as explored digitally as small-molecules may be encountered (e.g. polymers) in some mixtures but not in others. In CHEMIXHUB, a wide range of chemical species is present (see Figure 2) spanning salts and polymers. This diversity should be taken into account when modeling mixtures, the correct representation is still an open problem. All the chemical species in this datasets can be expressed as a SMILES string, which are standardized using RDKit. Polymers are represented by their monomeric units, along with their Mass average molar mass ($M_W$) or Number average molar mass ($M_N$). Molecules with ionic bonding (salts) are preserved and flagged in the dataset.

- **Handling of chemical 3D geometry** In the datasets we consider, the majority of molecules possess fewer than five rotatable bonds (see Appendix A.2). This limits the expected benefit of conformer ensemble approaches in our context [82]. We leave the study of the impact of 3D conformations information on property prediction of mixtures of highly flexible molecules for future work.

- **Number of components** While our focus is on multi-component systems, we preserve the single-component data points in the datasets with the exception of IlThermo (see Figure 2). We leave the choice to the user to filter out single-component data points in CHEMIXHUB.

- **Representing mixture composition** All possible compositional ratio were converted to mole fractions, discarding data points that did not have the information to make that conversion.

- **Missing temperature values** Standard conditions (298.15K) are assumed if temperature values are not reported. For the datasets where this is the case, added values are flagged. This flag can optionally be passed to the model for it to implicitly learn uncertainty over those assumed values.

- **Data scales** Due to great variations in experimental value ranges, it is common to apply logarithmic scaling to conductivity, solubility and viscosity properties [6, 7, 3]. We follow this principle and apply it to these types of properties.

## 3.4 Dataset splitting strategies

Aside from traditional random cross-validation (CV) splits with a default of 5-fold 70/10/20 training/validation/test splits, we propose 3 additional splitting strategies for benchmarking, to explore generalization capabilities of models:

- **Mixture size splits**: For a given threshold, the training set only contains mixtures with components that have a number of components less than the threshold, and the test set contains only mixtures that are above the threshold. For the olfactory similarity task, we employ the geometric mean of the two mixtures. This setting is interesting in industry because we want to predict the properties of complex mixtures while training on simpler, cheaper ones.

- **Leave-molecules-out (LMO) splits**: The test sets are split from the dataset such that certain molecules will not appear in the training set. Studying new molecules is an important consideration when validating models to ensure the model is applicable in out of distribution molecular discovery settings.

- **Temperature splits**: As highlighted in Table 1, multiple tasks in CHEMIXHUB have a temperature dependency. It is also desirable for industry to be able to predict the properties of certain mixtures in different temperature ranges than the training ones. We bin the temperature range into 5 categories based on the temperature distribution observed across the 11 tasks and use the bins as a 5-fold split.

## 4 Benchmark

### 4.1 Modeling space

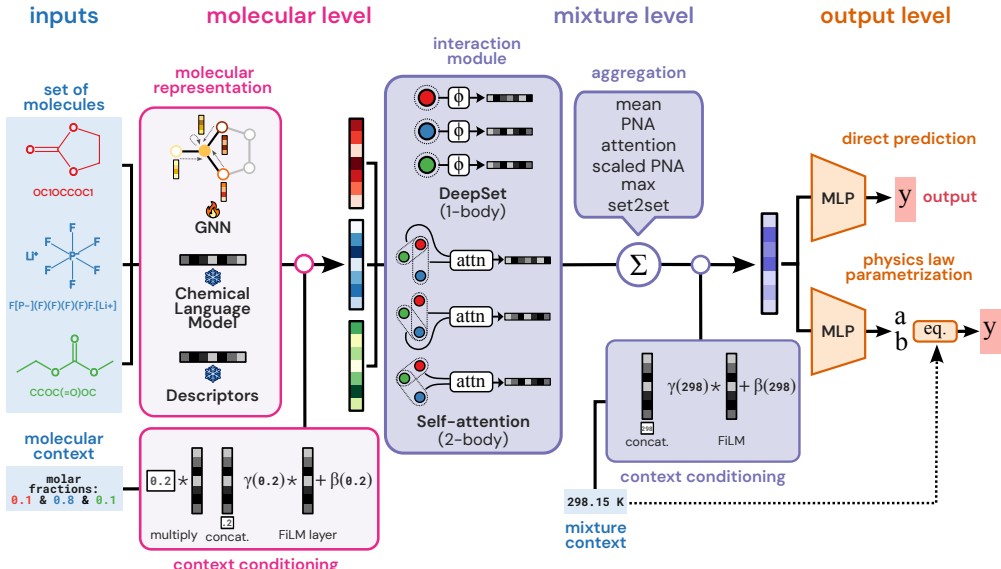

Figure 3: **Mapping out the deep learning modeling space for chemical mixtures**. We highlight three levels: (1) molecular representation and context infusion (e.g., molecular fraction), (2) mixture-level interaction aggregation, and infusion of global mixture context (e.g., temperature), (3) property output generation, each offering distinct avenues for model development.

The inherent set-like structure of molecular mixtures mandates that deep learning models incorporate specific inductive biases. Primarily, models must ensure permutation invariance, meaning the predicted property remains unaffected by the input order of constituent molecules and their compositions. Additionally, they must be flexible to a varying number of input components, allowing, for example, a model trained on binary mixtures to generalize to ternary systems or more complex formulations. These symmetries are critical for downstream applications where industrial formulations rely on precise compositions, ingredients are combined without a fixed order, and the effects of component addition or removal are routinely assessed.

To guide model development, we define a structured modeling space (Figure 3) that operates on input mixture data. Each data point comprises a set of pure component molecules, their associated molecular context (e.g., mole fractions), and the overall mixture context (e.g., temperature). This space, while focusing on foundational one-body and pairwise interactions and various aggregation operations, is non-exhaustive; future work could explore explicit N-body interactions or more sophisticated set-to-vector encoding mechanisms. Conceptually, this modeling space is divided into three levels: 1) molecular representation, 2) mixture representation, and 3) output generation.

**Molecular representation level** We benchmark three common embedding techniques: GNNs, CLMs and molecular descriptors. For GNNs, we use a GRAPHNETS architecture [4] trained end-to-end with the other module to learn the embeddings during the task (details in Section A.4), while for CLMs we rely on frozen pre-trained representations from MOLT5 [18]. All training is performed using the Adam optimizer [30]. Cheminformatics molecular descriptors are normalized 200 dimensional from RDKIT obtained from DESCRIPTASTORUS [29]. After obtaining the molecular embeddings, we consider three different ways of infusing the molecular context into them: 1) element-wise multiplication, 2) concatenation, 3) feature-wise linear modulation (FiLM) layer [43].

**Mixture representation level** We explore two possible interaction modules: DeepSets [73] and self-attention [61], which can be thought of as enabling one-body and two-body interactions between molecules in the mixture, respectively. We consider six different types of permutation invariant aggregation operations: mean, max, attention-based aggregation [61], principal neighborhood aggregation (PNA) [16], PNA scaled according to the number of component in the mixture and set2set [64] which yields the mixture embedding. We then consider two different ways of incorporating the mixture context into it: 1) concatenation 2) using a FiLM layer.

**Output level** We consider using either a fully-connected predictive head that directly outputs a predicted value, or for tasks that are known to be modeled by an Arrhenius relationship, predicting the parameters of the Arrhenius equation and using it to determine the final outputted value (see Section 4.4).

**Baseline** To establish a strong non-deep learning baseline, we provide comparison with a gradient-boosted random forest model, XGBoost [12] using RDKIT descriptors or MOLT5 embeddings molecular features. These features are linearly combined with their respective molecular context and then concatenated with the overall mixture context to form the input for XGBoost (details in Section A.8).

### 4.2 Performances across tasks

Table 2: **Model performances across CHEMIXHUB tasks** Reported MAE ($\downarrow$) on 5-fold random CV splits. The mean and standard deviation are reported.

| Molecular rep. | Mixture rep. | Miscible Solvents | | | Drug Solubility | SPE | NIST-full |
|---|---|---|---|---|---|---|---|
| | | $\rho$ | $\Delta H_{\mathrm{mix}}$ | $\Delta H_{\mathrm{vap}}$ | $\ln(S)$ | $\ln(\kappa)$ | $\ln(\eta)$ |
| GNN | Attention | 0.018 ± 0.020 | 0.158 ± 0.002 | 0.098 ± 0.006 | 0.087 ± 0.006 | 0.312 ± 0.043 | 0.136 ± 0.010 |
| | Deepsets | **0.003 ± 0.000** | 0.159 ± 0.002 | 0.406 ± 0.668 | 0.065 ± 0.005 | 0.315 ± 0.067 | 0.131 ± 0.010 |
| MolT5 | XGB | 0.009 ± 0.000 | 0.269 ± 0.004 | 0.306 ± 0.003 | **0.028 ± 0.001** | **0.222 ± 0.007** | 0.148 ± 0.001 |
| | Attention | 0.005 ± 0.001 | **0.157 ± 0.002** | 0.125 ± 0.077 | 0.082 ± 0.027 | 0.279 ± 0.006 | 0.076 ± 0.004 |
| | Deepsets | 0.008 ± 0.005 | 0.157 ± 0.003 | **0.071 ± 0.002** | 0.130 ± 0.013 | 0.328 ± 0.010 | 0.162 ± 0.009 |
| RDKit | XGB | 0.009 ± 0.000 | 0.225 ± 0.005 | 0.295 ± 0.002 | **0.028 ± 0.001** | 0.223 ± 0.008 | **0.055 ± 0.000** |
| | Attention | 0.006 ± 0.001 | 0.167 ± 0.002 | 0.199 ± 0.030 | 0.070 ± 0.006 | 0.394 ± 0.028 | 0.069 ± 0.006 |
| | Deepsets | 0.005 ± 0.000 | 0.207 ± 0.008 | 0.079 ± 0.005 | 0.179 ± 0.011 | 0.344 ± 0.016 | 0.137 ± 0.005 |

| Molecular rep. | Mixture rep. | IlThermo | | Fuel mixtures | NIST | Olfactory mixtures |
|---|---|---|---|---|---|---|
| | | $\ln(\kappa)$ | $\ln(\eta)$ | MON | $\ln(\eta)$ | Perceptual similarity |
| GNN | Attention | 0.276 ± 0.044 | 0.154 ± 0.084 | 10.240 ± 1.658 | 0.035 ± 0.004 | 0.129 ± 0.005 |
| | Deepsets | 0.226 ± 0.017 | 0.206 ± 0.020 | 5.990 ± 1.382 | 0.091 ± 0.005 | 0.146 ± 0.010 |
| MolT5 | XGB | **0.071 ± 0.001** | 0.078 ± 0.002 | 5.002 ± 0.538 | 0.059 ± 0.001 | 0.128 ± 0.006 |
| | Attention | 0.244 ± 0.011 | 0.083 ± 0.035 | 4.660 ± 0.603 | **0.030 ± 0.001** | 0.123 ± 0.005 |
| | Deepsets | 0.196 ± 0.013 | 0.132 ± 0.003 | 5.296 ± 0.585 | 0.056 ± 0.004 | **0.121 ± 0.006** |
| RDKit | XGB | 0.073 ± 0.002 | **0.076 ± 0.002** | **4.570 ± 0.348** | 0.048 ± 0.002 | 0.125 ± 0.006 |
| | Attention | 0.407 ± 0.019 | 0.100 ± 0.003 | 11.297 ± 2.110 | 0.056 ± 0.004 | 0.148 ± 0.010 |
| | Deepsets | 0.290 ± 0.059 | 0.107 ± 0.007 | 7.625 ± 1.874 | 0.047 ± 0.003 | 0.150 ± 0.008 |

We investigate how the architectural considerations highlighted in Section 4.1 impact the predictive power of models across the 11 tasks in CHEMIXHUB. We first select the best performing architectures that covers all combinations of molecular representation and mixture interaction modules (6 models) by a Bayesian optimization hyperparameter search (details in section A.7) on each task. Then, we train and test the selected model on 5-fold random CV splits (70/10/20 training/validation/test split). We report results across mean absolute error (MAE). The results compiled from the CV splits for all models evaluated are tabulated in Table 2. Additional metrics (Pearson correlation coefficient $\rho$ and Kendall ranking coefficient $\tau$) are reported in Section A.9.

We observe that traditional tree-based methods like XGBOOST are robust baselines on a great variety of tasks. It is interesting to note that XGBOOST-based methods greatly struggle on predicting $\Delta H_{\mathrm{mix}}$ and $\Delta H_{\mathrm{vap}}$, two properties well known for their non-linear mixture behavior. Overall, we observe pre-trained representations of CLMs like MOLT5 tend to yield better performances across our dataset compared to GNN-based representation and cheminformatics descriptors. We believe that pre-training on related data would greatly improve the performances of GNNs, as it has been shown to in the literature [59, 55]. Regarding the choice of mixture interaction module, the need for higher level of interactions tend to be task-dependent, with no consistent advantage of one method over the other.

### 4.3    Generalization to new mixture sizes and molecules

We further study how robust models are to variation in the number of components in mixtures and to new molecular entities. For the scope of this study, we focus on the datasets which have the greatest variation in terms of number of components, namely the MON and Olfactory similarity datasets. We assess the performance of the best deep learning model for each task as determined in Section 4.2 and report it using the Pearson correlation coefficient $\rho$ in Figure 4.

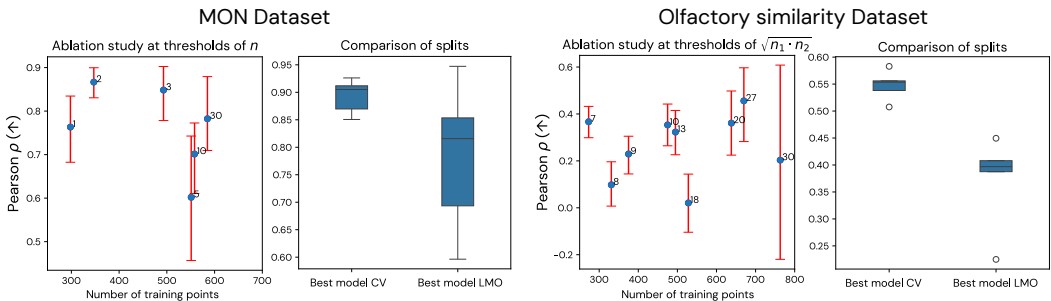

Figure 4: **Generalization to new mixture sizes and molecules**. For each dataset: (Left) Ablation study with training data only containing mixtures with (geometric) average number of molecules less than a threshold. The thresholds are indicated for each split. (Right) Boxplot of the best deep learning model test Pearson correlation on random CV splits, and the LMO splits.

We observe great sensitivity to mixture sizes across both datasets. We hypothesize that addition of new datapoints with slightly higher numbers of components may be considered as noise by the model and therefore leads to overfitting. Additionally, we observe a significant decrease in performance when considering new chemistries. This lack of extrapolative behavior regarding individual molecular species is expected, as we noted that the molecular level modeling plays an essential part in model performance.

### 4.4    Explicit physics-based modeling improves performances

Previous work have reported that incorporating known physical or chemical constraints into ML models can improve accuracy and generalizability of model predictions [7, 81]. We investigate how swapping a regular fully connected predictive head for a "physics-based" predictive head impacts the model's performance. Namely, we modify the predictive head to output the coefficients of a physics law suitable for the task. For the scope of this paper, we limit our study to temperature-dependent tasks whose target property can be effectively modeled by the Arrhenius equation.

$$\ln(y) = \ln(A) - \frac{E_a}{RT} \tag{1}$$

where y can be viscosity $\eta$ and ionic conductivity $\kappa$ tasks, $R$ is the perfect gas law constant, $T$ is the given temperature and $A$ and $E_a$ parameters to predict. We assess the performance of the

best deep learning model for each task, as determined in Section 4.2 and conduct the study on temperature dependent splits to investigate generalizability to new temperature ranges. We also report the performance of XGBOOST on this harder type of split.

Table 3: **Physics bias improves performances across temperature-dependent tasks** Reported on temperature range exclusion splits, up to 5-fold. The mean and standard deviation are reported.

| Metric | Model | IlThermo | | NIST | SPE |
|---|---|---|---|---|---|
| | | $\ln(\kappa)$ | $\ln(\eta)$ | $\ln(\eta)$ | $\ln(\kappa)$ |
| MAE ($\downarrow$) | Best XGB | 0.386 ± 0.142 | 0.432 ± 0.160 | 0.126 ± 0.022 | 0.482 ± 0.138 |
| | Best model | 0.354 ± 0.152 | 0.162 ± 0.096 | 0.079 ± 0.006 | 0.481 ± 0.228 |
| | Best model + Arrhenius | **0.284 ± 0.025** | **0.127 ± 0.011** | **0.048 ± 0.013** | **0.363 ± 0.015** |
| Pearson $\rho$ ($\uparrow$) | Best XGB | 0.940 ± 0.056 | 0.941 ± 0.063 | 0.877 ± 0.008 | 0.935 ± 0.018 |
| | Best model | 0.923 ± 0.084 | 0.968 ± 0.033 | 0.946 ± 0.006 | 0.941 ± 0.024 |
| | Best model + Arrhenius | **0.987 ± 0.002** | **0.988 ± 0.003** | **0.980 ± 0.011** | **0.961 ± 0.003** |

We observe that adding a physics bias via the Arrhenius equation greatly improves the performance of deep learning architectures in this setting. This technique also allows better interpretability of the predictive model, as it grounds it in known equations. Additionally, we note that the performance of XGBOOST, a model that typically performs well on randomized splits, has significantly decreased performance when evaluated on a harder and more realistic split, highlighting the importance of generalizability assessment and not relying on a single split toi report in literature.

### 4.5 Transfer learning capabilities of models within the Miscible Solvent dataset tasks.

We evaluate transfer learning capabilities of the best performing models on each of the 3 property prediction tasks (Density $\rho$, $\Delta H_{\mathrm{mix}}$ and $\Delta H_{\mathrm{vap}}$) of the Miscible Solvents dataset. For each task, we finetune the best models of the other two tasks and compare their performance to the original best model for this task reported in Section 4.2.

Table 4: **Intra-dataset transfer learning capabilities depend on task difficulty** Metrics are reported on 5-fold random CV splits. The mean and standard deviation are reported. The original best model statistics are taken from Section 4.2 and Appendix A.9.

| Fine-tuning Dataset | Best model Original Dataset | Pearson $\rho$ ($\uparrow$) | MAE ($\downarrow$) | Kendall $\tau$ ($\uparrow$) |
|---|---|---|---|---|
| $\rho$ | $\rho$ | **0.999 ± 0.000** | **0.003 ± 0.000** | **0.973 ± 0.000** |
| | $\Delta H_{\mathrm{mix}}$ | 0.955 ± 0.006 | 0.021 ± 0.001 | 0.824 ± 0.009 |
| | $\Delta H_{\mathrm{vap}}$ | 0.929 ± 0.008 | 0.026 ± 0.002 | 0.769 ± 0.018 |
| $\Delta H_{\mathrm{vap}}$ | $\Delta H_{\mathrm{vap}}$ | **0.999 ± 0.000** | **0.071 ± 0.002** | **0.976 ± 0.001** |
| | $\Delta H_{\mathrm{mix}}$ | 0.808 ± 0.017 | 1.063 ± 0.057 | 0.611 ± 0.034 |
| | $\rho$ | 0.644 ± 0.088 | 1.366 ± 0.176 | 0.465 ± 0.074 |
| $\Delta H_{\mathrm{mix}}$ | $\Delta H_{\mathrm{mix}}$ | **0.976 ± 0.003** | **0.157 ± 0.002** | **0.835 ± 0.002** |
| | $\Delta H_{\mathrm{vap}}$ | 0.626 ± 0.022 | 0.527 ± 0.008 | 0.439 ± 0.025 |
| | $\rho$ | 0.348 ± 0.044 | 0.629 ± 0.013 | 0.237 ± 0.033 |

We observe dramatic differences depending on the task the model was originally trained on: Models initially trained to predict highly non-linear properties — harder tasks — like $\Delta H_{\mathrm{mix}}$ and $\Delta H_{\mathrm{vap}}$ perform really well when finetuned to predict density $\rho$ but the model initially trained to predict density $\rho$ fails at delivering good performances on $\Delta H_{\mathrm{mix}}$ and $\Delta H_{\mathrm{vap}}$ predictions. Architectural differences may also play a role in this phenomenon. We perform additional inter-dataset transfer learning experiments in Appendix A.10.

### 4.6 Zero-shot capabilities across viscosity prediction tasks

We performed additional experiments to investigate the zero-shot capabilities of the best performing deep learning models for each of the viscosity ($\ln(\eta)$) prediction tasks in CheMixHub and observe good zero shot capabilities for tasks that have similar viscosity value ranges.

Table 5: **Zero shot learning capabilities of models across the Viscosity** $\ln(\eta)$ **prediction tasks in CheMixHub** Metrics are reported on 5-fold random CV splits. The mean and standard deviation are reported. The original best model statistics are taken from Section 4.2 and Appendix A.9.

| Zero-shot Dataset | Best model Original Dataset | Pearson $\rho$ ($\uparrow$) | MAE ($\downarrow$) | Kendall $\tau$ ($\uparrow$) |
|---|---|---|---|---|
| NIST | NIST | **0.991 ± 0.001** | **0.030 ± 0.001** | **0.939 ± 0.001** |
| | NIST-full | 0.985 ± 0.002 | 6.806 ± 0.012 | 0.926 ± 0.004 |
| | IlThermo $\ln(\eta)$ | 0.575 ± 0.028 | 5.880 ± 0.129 | 0.451 ± 0.024 |
| NIST-full | NIST-full | **0.992 ± 0.000** | **0.055 ± 0.000** | 0.966 ± 0.000 |
| | IlThermo $\ln(\eta)$ | 0.775 ± 0.018 | 0.811 ± 0.078 | **1.000 ± 0.000** |
| | NIST | 0.694 ± 0.021 | 6.281 ± 0.005 | **1.000 ± 0.000** |
| IlThermo $\ln(\eta)$ | IlThermo $\ln(\eta)$ | **0.995 ± 0.001** | **0.076 ± 0.002** | **0.968 ± 0.001** |
| | NIST-full | 0.956 ± 0.004 | 0.276 ± 0.032 | 0.880 ± 0.015 |
| | NIST | 0.452 ± 0.041 | 4.815 ± 0.030 | 0.330 ± 0.047 |

## 5  Conclusion

In this study, we introduced CHEMIXHUB, a comprehensive suite of datasets and benchmarks designed to accelerate research in chemical mixture property prediction. Addressing the critical need for standardized resources in a field characterized by scattered datasets, inconsistent evaluation protocols, and limited open-source model implementations, CHEMIXHUB provides a curated collection of 11 tasks, diverse splitting strategies for robust generalization assessment, and initial baselines using representative ML models. Our work aims to lower the barrier to entry and foster systematic progress in understanding and modeling these complex multi-molecular systems. Our benchmarking revealed that traditional models like XGBoost with appropriate chemical features offer strong baselines on random splits, often rivaling more complex deep learning methods. This highlights the necessity for deep learning approaches to demonstrate clear advantages, particularly on more challenging out-of-distribution tasks. We observed that datasets with greater monomolecular diversity (e.g., fuel and olfactory mixtures) benefit from hierarchical modeling as well as tasks with well known non-linear relationships, such as enthalpy of mixing $\Delta H_{\mathrm{mix}}$ and heat of vaporization $\Delta H_{\mathrm{mix}}$, underscoring the need for advanced modeling and rigorous evaluation beyond simple random splits. Encouragingly, the explicit incorporation of physics-based constraints, like the Arrhenius equation for temperature-dependent properties, significantly enhanced model performance and generalization, suggesting a fruitful direction for future work in fusing domain knowledge with data-driven techniques. The optimal level of interaction modeling—whether one-body (*DeepSets*-like) or explicit many-body approaches—also remains task-dependent and warrants further investigation, alongside innovations in aggregation, context conditioning, and attention mechanisms tailored for mixtures. CHEMIXHUB is intended to catalyze progress across these diverse research frontiers, equipping the community to tackle the complex and impactful domain of chemical mixture modeling for better drugs and materials. We condemn any malicious use of our work to create malicious or hazardous chemicals.

## 6  Limitations

Several limitations of current approaches and avenues for future research are illuminated by CHEMIXHUB. Representing complex entities like polymers, currently simplified to monomeric units, requires more sophisticated featurization. Beyond property prediction, the vast chemical mixture space invites exploration into formulation discovery, optimization, and de novo design. These endeavors, especially those involving iterative experimental design, are nascent and present significant ML challenges, particularly given the often data-scarce nature of experimental mixture datasets. Thus, techniques for data-efficient learning, multi-task approaches, and robust pre-training strategies are crucial. While the GNNs in our study were not pre-trained, exploring task-specific or general pre-training for mixture-aware GNNs or CLMs is a promising direction. Finally, enhancing model interpretability—providing insights at both molecular and mixture interaction levels—is essential for the practical adoption of these models in chemical research and industry.

## Acknowledgments and Disclosure of Funding

E. M. R. and B. S.-L. would like to thank Prof. Alán Aspuru-Guzik for his support and advice. This research was enabled in part by computational resources provided by the Digital Research Alliance of Canada (https://alliancecan.ca) and the Acceleration Consortium (https://acceleration.utoronto.ca). The authors gratefully acknowledge financial support from the Acceleration Consortium, the Natural Sciences and Engineering Council of Canada (NSERC), University of Toronto's Data Science Institute and the Vector Institute.

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

# A  Appendix

## A.1  Dataset licenses

We list the different licenses of the dataset curated in CHEMIXHUB below:

- Miscible Solvent: CC BY-NC 4.0
- IlThermo: CC BY 4.0
- NIST TRC SOURCE Zenodo archive: CC BY 4.0
- Drug solubility: CC BY 4.0
- Solid polymer electrolyte: MIT
- Motor Octane Number: CC BY 4.0
- Olfactory Similarity: CC BY 4.0

## A.2  Additional statistics on molecules for each of the 11 tasks in CheMixHub

Table 6: **Additional statistics on molecules for each of the 11 tasks in CheMixHub**.

| Dataset | Tasks | Avg. # Atoms/Mol | Max # Atoms/Mol | Min # Atoms/Mol | Avg # Fragments | Max # Fragments | Avg Molecular Weight | Avg # Rotatable Bonds | Avg Components Mixture |
|---|---|---|---|---|---|---|---|---|---|
| Miscible solvents | $\rho$ $\Delta H_{\text{mix}}$ $\Delta H_{\text{vap}}$ | 8.28±3.17 | 18 | 3 | 1.0±0.0 | 1 | 123.73±43.96 | 3.40±3.17 | 3.72±1.08 |
| IlThermo | $\ln(\kappa)$ $\ln(\eta)$ | 15.80±9.28 17.33±10.73 | 77 62 | 1 1 | 1.76±0.54 1.85±0.59 | 4 4 | 250.91±145.56 280.30±174.57 | 5.12±6.13 5.76±6.51 | 2.21±0.41 2.40±0.49 |
| NIST Viscosity | $\ln(\eta_{\text{NIST-full}})$ $\ln(\eta_{\text{NIST}})$ | 12.90±8.98 9.12±4.71 | 95 63 | 1 1 | 1.50±0.70 1.0±0.0 | 8 1 | 203.98±135.28 140.52±73.17 | 4.00±5.60 3.14±3.79 | 1.88±0.33 1.92±0.28 |
| Drug solubility | $\ln(S)$ | 14.48±9.16 | 51 | 1 | 1.11±0.33 | 3 | 212.40±128.17 | 2.37±2.45 | 1.91±0.29 |
| Solid Polymer Electrolyte | $\ln(\kappa)$ | 30.86±47.75 | 676 | 2 | 1.24±0.44 | 3 | 473.36±738.30 | 18.11±33.19 | 2.24±0.67 |
| Olfactory mixtures | Perceptual similarity | 9.53±3.43 | 21 | 3 | 1.0±0.0 | 1 | 135.67±45.03 | 2.72±2.29 | 13.30±10.51 |
| Fuel mixtures | MON | 7.93±1.94 | 12 | 2 | 1.0±0.0 | 1 | 110.66±26.19 | 1.71±1.69 | 5.69±14.24 |

## A.3  IlThermo Dataset curation details

We use the ILTHERMOPY package to retrieve IlThermo entries, selecting entries that are either binary or ternary mixtures and corresponding to our property of choice (for the scope of this paper, we limit ourselves to viscosity and ionic conductivity properties) [26]. We remove mixture that exhibits multiple phases behavior and are not liquid at the indicated temperature. We apply a natural logarithm transformation to the viscosity and ionic conductivity values present in IlThermo to make the range of values easier to learn. We also constrain the pressure range to be near the standard value of 1 atm or 101.325 kPa by applying a ±2kPa threshold on pressure values.

We then standardize the mixture composition metric to mole fraction by converting as many entries as possible into that format. Data points which have the mixture composition expressed using molarity are discarded, as the conversion would require making assumption about the component densities.

Assuming a binary mixture of component $A$ and $B$ with a given mole ratio $r_{A:B} = \frac{n_A}{n_B}$ where $n_A$ and $n_B$ are the number of moles of $A$ and $B$, respectively, the mole fractions $\chi_A$ and $\chi_B$ can be calculated using:

$$\chi_A = \frac{n_A}{n_A + n_B} = \frac{r_{A:B}}{r_{A:B} + 1} \tag{2}$$

$$\chi_B = 1 - \chi_A \tag{3}$$

Similarly, assuming a ternary mixture of component $A$, $B$ and $C$ with given mole ratios $r_{A:B} = \frac{n_A}{n_B}$ and $r_{A:C} = \frac{n_A}{n_C}$, the mole fractions $\chi_A$, $\chi_B$ and $\chi_C$ can be retrieved using:

$$\chi_A = \frac{n_A}{n_A + n_B + n_C} = \frac{r_{A:B}}{r_{A:B} + \frac{r_{A:B}}{r_{A:C}} + 1} \tag{4}$$

$$\chi_B = \frac{n_B}{n_A + n_B + n_C} = \frac{\frac{1}{r_{A:B}}}{\frac{1}{r_{A:B}} + \frac{1}{r_{A:C}} + 1} \tag{5}$$

$$\chi_C = \frac{\frac{1}{r_{A:C}}}{\frac{1}{r_{A:B}} + \frac{1}{r_{A:C}} + 1} \tag{6}$$

Assuming a binary mixture of component $A$ and $B$, and given the mass ratio $r_{A:B} = \frac{m_A}{m_B}$ where $m_A$ and $m_B$ are the mass of $A$ and $B$ in g, respectively and the molecular weights $MW_A$ and $MW_B$, to retrieve the mole fractions $\chi_A$ and $\chi_B$, we first calculate mass fractions $\gamma_A$ and $\gamma_B$ using:

$$\gamma_A = \frac{m_a}{m_a + m_b} = \frac{r_{A:B}}{r_{A:B} + 1} \tag{7}$$

$$\gamma_B = 1 - \gamma_A \tag{8}$$

then assuming $m_{tot} = m_A + m_B = 1$g, we use $m_A = \gamma_A m_{tot}$ and $m_B = \gamma_B m_{tot}$ to obtain

$$n_A = \frac{m_A}{MW_A} \tag{9}$$

$$n_B = \frac{m_B}{MW_B} \tag{10}$$

$$n_{tot} = n_A + n_B \tag{11}$$

$$\chi_A = \frac{n_A}{n_{tot}} \tag{12}$$

$$\chi_B = 1 - \chi_A \tag{13}$$

The same process is naturally extended for ternary mixtures, assuming $r_{C:B} = \frac{m_C}{m_B}$ and $MW_C$ are given.

Assuming a binary mixture of component $A$ and $B$, and given the molarity $M_A = \frac{n_A}{m_B}$ where $m_B$ is the mass of $B$ in kg and $n_A$ the number of moles of $A$ and the molecular weights $M_A$ and $M_B$, to retrieve the mole fractions $\chi_A$ and $\chi_B$, we assume $m_B = 1$kg so $M_A = n_A$ and use $n_B = \frac{m_B}{MW_B}$ to obtain:

$$\chi_A = \frac{n_A}{n_A + n_B} = \frac{M_A}{M_A + \frac{1000}{MW_B}} \tag{14}$$

$$\chi_B = 1 - \chi_A \tag{15}$$

where a factor of 1000 is introduced since $M_A$ and $M_B$ are expressed in g/mol. The same process is naturally extended for ternary mixtures, assuming $M_C = \frac{n_C}{m_B}$ and $MW_C$ are given.

Assuming a binary mixture of component $A$ and $B$, and given the weight fraction $\gamma_A$ and the molecular weights $M_A$ and $M_B$, to retrieve the mole fractions $\chi_A$ and $\chi_B$, we assume $m_{tot} = m_A + m_B = 1$g and use $m_A = \gamma_A m_{tot}$ and $m_B = \gamma_B m_{tot}$ to obtain:

$$n_A = \frac{m_A}{MW_A} = \frac{\gamma_A}{MW_A} \tag{16}$$

$$n_B = \frac{m_B}{MW_B} = \frac{\gamma_B}{MW_B} = \frac{1 - \gamma_A}{MW_B} \tag{17}$$

$$\chi_A = \frac{n_A}{n_A + n_B} \tag{18}$$

$$\chi_B = 1 - \chi_A \tag{19}$$

The same process is naturally extended for ternary mixtures, assuming $\gamma_C$ and $MW_C$ are given.

### A.4 Details of molecular graph representation

The GNN takes in molecular graphs derived from the SMILES representations of molecules. Each graph, written as $G = (U, V, E)$, consists of a special global vertex $U$ connected to all other vertices $V$, and a set of edges $E$. The global vertex $U$ encodes overall properties of the molecule and is initialized with 200 normalized RDKIT descriptors obtained from DESCRIPTASTORUS [29]. The atoms of the molecules are the vertices (nodes), with node vectors $V = \{v_i\}_{i=1}^{N_v}$ for a molecule with $N_v$ atoms, where $v_i$ are feature vectors encoding atomic properties. Covalent bonds between atoms are represented as edges $E = \{(e_k, r_k, s_k)\}_{k=1}^{N_e}$ for a molecule with $N_e$ bonds, where $e_k$ are feature vectors of edge properties, and $r_k, s_k \in [1, \ldots, N_v]$ are indices of the two atoms that the bond joins together. Note $r_k \neq s_k$, since bonds must be between two different atoms

The node features used in the molecular graph representation as input to the GNN are 85-dimensional one-hot encoding vectors, encoding categorical information about the atoms. The edge features encode the categorical information about the bonds as 14-dimensional one-hot encoding vectors. The molecular information for the features are shown in Table 7.

Table 7: Features for node and edge features of molecular graphs

All categories are one-hot encoded and stacked to give a singular bit vector. UNK stands for "unknown", and is a catch-all category.

| Node features | Categories |
| --- | --- |
| Atomic number | 1 (hydrogen) to 54 (iodine), UNK |
| Atom degree | 0, 1, 2, 3, 4, 5, UNK |
| Formal charge | -2, -1, 0, 1, 2, UNK |
| Chirality | unspecified, CW, CCW, other, UNK |
| Number of hydrogens | 0, 1, 2, 3, 4, 5, 6, 7, 8, UNK |
| Hybridization | sp, sp2, sp3, sp3d, sp3d2, UNK |
| Aromatic | True/False |

| Edge features | Categories |
| --- | --- |
| Bond type | single, double, triple, aromatic, UNK |
| Is conjugated | True/False |
| In ring | True/False |
| Stereo-configuration | none, $Z$, $E$, *cis*, *trans*, any, UNK |

As mentioned in Section 3.3, polymers and salts are present in the dataset and this probes important modeling considerations when employing GNNs. For polymers, we decided to restrict our modeling consideration to passing their monomeric units to the GNN. For salts, we conducted a chemical analysis to determine the impact of modeling the cation and anion as one disconnected graph. The details of it can be found in Section A.12.

### A.5 Compute resources details

All model training/validation was conducted on a single A100 40GB NVDIA GPU.

### A.6 Training details

Each run was performed for 500 epochs using the Adam optimizer [30], with a batch size set to 1024. Early stopping was implemented with patience set to 100. Two different learning rates were used to train the models end-to-end, one for the molecular-level model and one for the rest of the model. The splits used are specified in Section 3.4, further details on hyperparameter tuning can be found in Section A.7.

### A.7 Hyperparameter search

For each task, the search was performed using Weights & Biases [5] with the BOHB algorithm [20] and a budget of 160 runs. 80 runs were allocated to the GNN-based molecular representations and 80 to CLMs and descriptors runs. Each run was performed for 500 epochs with early stopping patience set to 100. The search was conducted using the first split of the 5-fold random CV splits (70/10/20 training/validation/test split). The search space is defined as follows

- Molecular featurization: ["custom molecular graphs", "molt5 embeddings", "rdkit2d normalized features"]

- General hyper-parameters:
  - Loss type: ["mae", "mse"]
  - Dropout rate: [0, 0.05, 0.1, 0.15, 0.2, 0.25, 0.3, 0.35, 0.4, 0.45, 0.5]
  - Learning rate (molecular level): [8e-5, 5e-5, 1e-4, 5e-4, 8e-4, 1e-3, 5e-3, 1e-2]
  - Learning rate (mixture level and head): [8e-5, 5e-5, 1e-4, 5e-4, 8e-4, 1e-3, 5e-3, 1e-2]
- Molecular-level hyper-parameters:
  - Molecular context aggregation type: ["concatenate", "multiply", "film"]
  - FiLM layer activation function: ["sigmoid", "relu"]
- Mixture-level hyper-parameters:
  - Mixture interaction module: ["self attention", "deepset"]
  - MLP head in self-attention: ["True", "False"]
  - Embedding dimension: [32, 64, 96, 128]
  - Number of layers: [0, 1, 2, 3]
  - Aggregation type: ["mean", "max", "pna", "scaled pna", "attention", "set2set"]
  - Number of attention heads: [1, 4, 8, 16]
  - Output dimension: [96, 128, 256]
  - Mixture context aggregation type: ["concatenate", "film"]
  - FiLM layer activation function (mixture context): ["sigmoid", "relu"]
- Predictive head hyper-parameters:
  - Embedding dimension: [64, 128, 192, 256, 320]
  - Number of layers: [1, 2, 3]

For runs where the molecular featurization used GNNs, the following additional parameters were added to the search space:

- GNN hyper-parameters:
  - Embedding dimension: [64, 128, 192, 256, 320]
  - Number of layers: [2, 3, 4]

### A.8 XGBOOST modeling

The XGBOOST model was given a maximum of 1,000 estimators and tree depth of 1,000 except for the NIST-full task, where a maximum of 250 estimators and a tree depth of 250 was used. To ensure the model does not overfit, we use the validation set for early stopping, with a patience of 25 epochs. The model is trained with mean squared error, with a learning rate of 0.01.

In addition to the MAE results reported in Section 4.2, we report the results compiled from the CV splits for all models evaluated in terms of Pearson correlation coefficient $\rho$ and Kendall ranking coefficient $\tau$ in Table 8 and 9, respectively.

Table 8: **Model performances across CHEMIXHUB tasks** Reported Pearson correlation coefficient $\rho$ ($\uparrow$) on 5-fold random CV splits. The mean and standard deviation are reported.

| Molecular rep. | Mixture rep. | Miscible Solvents | | | Drug Solubility | SPE | NIST-full |
|---|---|---|---|---|---|---|---|
| | | $\rho$ | $\Delta H_{\mathrm{mix}}$ | $\Delta H_{\mathrm{vap}}$ | $\ln(S)$ | $\ln(\kappa)$ | $\ln(\eta)$ |
| GNN | Attention | 0.948 ± 0.076 | **0.974 ± 0.003** | 0.998 ± 0.000 | 0.993 ± 0.001 | 0.970 ± 0.004 | 0.980 ± 0.002 |
| | Deepsets | **0.999 ± 0.000** | 0.974 ± 0.004 | 0.851 ± 0.296 | 0.996 ± 0.001 | 0.969 ± 0.010 | 0.981 ± 0.002 |
| MolT5 | XGB | 0.992 ± 0.001 | 0.924 ± 0.005 | 0.987 ± 0.001 | **0.999 ± 0.000** | 0.976 ± 0.001 | 0.989 ± 0.001 |
| | Attention | 0.998 ± 0.001 | 0.976 ± 0.003 | 0.997 ± 0.004 | 0.992 ± 0.005 | 0.973 ± 0.001 | 0.975 ± 0.038 |
| | Deepsets | 0.997 ± 0.001 | 0.976 ± 0.003 | **0.999 ± 0.000** | 0.983 ± 0.003 | 0.967 ± 0.002 | 0.977 ± 0.002 |
| RDKit | XGB | 0.992 ± 0.000 | 0.945 ± 0.007 | 0.986 ± 0.001 | **0.999 ± 0.000** | **0.977 ± 0.001** | **0.992 ± 0.000** |
| | Attention | 0.997 ± 0.001 | 0.972 ± 0.003 | 0.991 ± 0.003 | 0.996 ± 0.001 | 0.947 ± 0.008 | 0.995 ± 0.000 |
| | Deepsets | 0.996 ± 0.001 | 0.954 ± 0.005 | **0.999 ± 0.000** | 0.973 ± 0.004 | 0.963 ± 0.003 | 0.970 ± 0.002 |

| Molecular rep. | Mixture rep. | IlThermo | | MON | NIST | Olfaction |
|---|---|---|---|---|---|---|
| | | $\ln(\kappa)$ | $\ln(\eta)$ | MON | $\ln(\eta)$ | Mixture similarity |
| GNN | Attention | 0.986 ± 0.008 | 0.975 ± 0.012 | 0.360 ± 0.300 | 0.990 ± 0.002 | 0.447 ± 0.120 |
| | Deepsets | 0.989 ± 0.002 | 0.939 ± 0.060 | 0.820 ± 0.087 | 0.990 ± 0.002 | 0.132 ± 0.103 |
| MolT5 | XGB | **0.997 ± 0.000** | **0.995 ± 0.001** | 0.860 ± 0.032 | 0.950 ± 0.003 | 0.432 ± 0.047 |
| | Attention | 0.988 ± 0.001 | 0.993 ± 0.003 | 0.893 ± 0.028 | **0.991 ± 0.001** | **0.559 ± 0.040** |
| | Deepsets | 0.993 ± 0.002 | 0.986 ± 0.002 | 0.880 ± 0.036 | 0.981 ± 0.001 | 0.548 ± 0.025 |
| RDKit | XGB | **0.997 ± 0.001** | **0.995 ± 0.001** | **0.913 ± 0.019** | 0.957 ± 0.003 | 0.476 ± 0.062 |
| | Attention | 0.987 ± 0.003 | 0.981 ± 0.003 | 0.197 ± 0.351 | 0.977 ± 0.024 | 0.056 ± 0.130 |
| | Deepsets | 0.991 ± 0.002 | 0.992 ± 0.001 | 0.752 ± 0.155 | 0.980 ± 0.002 | -0.091 ± 0.050 |

Table 9: **Model performances across CHEMIXHUB tasks** Reported Kendall ranking coefficient $\tau$ ($\uparrow$) on 5-fold random CV splits. The mean and standard deviation are reported.

| Molecular rep. | Mixture rep. | Miscible Solvents | | | Drug Solubility | SPE | NIST-full |
|---|---|---|---|---|---|---|---|
| | | $\rho$ | $\Delta H_{\mathrm{mix}}$ | $\Delta H_{\mathrm{vap}}$ | $\ln(S)$ | $\ln(\kappa)$ | $\ln(\eta)$ |
| GNN | Attention | 0.910 ± 0.091 | 0.835 ± 0.004 | 0.969 ± 0.002 | 0.932 ± 0.003 | 0.868 ± 0.016 | 0.904 ± 0.007 |
| | Deepsets | **0.973 ± 0.000** | 0.833 ± 0.003 | 0.816 ± 0.318 | 0.949 ± 0.004 | 0.869 ± 0.023 | 0.905 ± 0.002 |
| MolT5 | XGB | 0.924 ± 0.00 | 0.730 ± 0.008 | 0.897 ± 0.003 | **0.978 ± 0.001** | **0.899 ± 0.003** | 0.950 ± 0.000 |
| | Attention | 0.963 ± 0.006 | **0.835 ± 0.002** | 0.955 ± 0.034 | 0.935 ± 0.022 | 0.881 ± 0.003 | 0.956 ± 0.003 |
| | Deepsets | 0.966 ± 0.002 | **0.835 ± 0.002** | **0.976 ± 0.001** | 0.893 ± 0.010 | 0.861 ± 0.004 | 0.910 ± 0.004 |
| RDKit | XGB | 0.929 ± 0.002 | 0.773 ± 0.005 | 0.898 ± 0.003 | **0.978 ± 0.001** | 0.899 ± 0.004 | **0.966 ± 0.000** |
| | Attention | 0.961 ± 0.003 | 0.829 ± 0.003 | 0.944 ± 0.012 | 0.948 ± 0.006 | 0.840 ± 0.014 | 0.957 ± 0.003 |
| | Deepsets | 0.956 ± 0.001 | 0.788 ± 0.008 | 0.973 ± 0.002 | 0.856 ± 0.009 | 0.855 ± 0.007 | 0.921 ± 0.002 |

| Molecular rep. | Mixture rep. | IlThermo | | MON | NIST | Olfaction |
|---|---|---|---|---|---|---|
| | | $\ln(\kappa)$ | $\ln(\eta)$ | MON | $\ln(\eta)$ | Mixture similarity |
| GNN | Attention | 0.923 ± 0.021 | 0.941 ± 0.029 | 0.348 ± 0.203 | 0.940 ± 0.007 | 0.312 ± 0.073 |
| | Deepsets | 0.930 ± 0.006 | 0.863 ± 0.103 | 0.687 ± 0.093 | **0.942 ± 0.009** | 0.166 ± 0.067 |
| MolT5 | XGB | **0.974 ± 0.000** | 0.967 ± 0.001 | 0.756 ± 0.038 | 0.863 ± 0.002 | 0.319 ± 0.047 |
| | Attention | 0.930 ± 0.004 | 0.967 ± 0.010 | 0.768 ± 0.033 | 0.939 ± 0.001 | 0.377 ± 0.042 |
| | Deepsets | 0.942 ± 0.004 | 0.945 ± 0.002 | 0.714 ± 0.012 | 0.896 ± 0.001 | **0.390 ± 0.011** |
| RDKit | XGB | 0.973 ± 0.001 | **0.968 ± 0.001** | **0.781 ± 0.029** | 0.883 ± 0.003 | 0.342 ± 0.040 |
| | Attention | 0.924 ± 0.015 | 0.957 ± 0.000 | 0.164 ± 0.266 | 0.916 ± 0.029 | 0.036 ± 0.065 |
| | Deepsets | 0.928 ± 0.005 | 0.956 ± 0.002 | 0.583 ± 0.121 | 0.897 ± 0.005 | -0.048 ± 0.035 |

## A.10   Additional transfer-learning benchmark

We evaluated transfer learning capabilities of two models trained on different datasets and tasks: the best deep learning model trained on the Miscible Solvent $\Delta H_{\mathrm{vap}}$ task and the other one trained on the Motor Octane Number (MON) task (according to Section 4.2). We compare these fine-tuned models to the best performing models for these tasks found in Section 4.2 (see Table 2).

We observe a simple fine-tuning approach of the best Deep Learning models for each task on another task from a different dataset does not yield good performance, especially compared to "in-dataset" finetuning results above, which could suggest the models are overfitting to their respective tasks.

An interesting experimental set up to further answer this questions would be to evaluate multi-task learning capabilities of these models across datasets, which should be easily implementable thanks to our unified framework.

Table 10: **Transfer learning capabilities of models across the Miscible Solvent (MS)** $\Delta H_{\text{vap}}$ **task and the MON task.** Metrics are reported on 5-fold random CV splits. The mean and standard deviation are reported. The best model statistics are taken from Section 4.2 and Appendix A.9.

| Fine-tuning Dataset | Best model Original Dataset | Pearson $\rho$ ($\uparrow$) | MAE ($\downarrow$) | Kendall $\tau$ ($\uparrow$) |
|---|---|---|---|---|
| MON | MON | 0.913 ± 0.019 | 4.570 ± 0.348 | 0.781 ± 0.029 |
| | MS-$\Delta H_{\text{vap}}$ | 0.160 ± 0.108 | 33.199 ± 1.606 | 0.144 ± 0.056 |
| Miscible Solvent $\Delta H_{\text{vap}}$ | MS-$\Delta H_{\text{vap}}$ | 0.999 ± 0.000 | 0.071 ± 0.002 | 0.976 ± 0.001 |
| | MON | 0.501 ± 0.095 | 1.582 ± 0.095 | 0.296 ± 0.067 |

## A.11 Additional benchmark

Table 11: **DiffMix tasks summary.** *T* indicates temperature dependency. *Mole Fractions* indicates mole fractions availability. *Arrhenius relationship* indicates if the task can be modeled using the Arrhenius equation. *Exp.* indicates if the data was obtained from wet-lab experiments or simulations.

| Tasks | | Units | Datapoints | Max # Components | # Unique Mixtures | # Unique Molecules | Mixture Context | Mole Fractions | Arrhenius Relationship | Exp. |
|---|---|---|---|---|---|---|---|---|---|---|
| DiffMix | $\kappa$ | mS/cm | 24,822 | 4 | 82 | 8 | T | ✓ | ✓ | ✗ |
| | $\Delta V$ | cm$^3$/mol | 1069 | 2 | 28 | 25 | T | ✓ | ✓ | ✓ |
| | $H_m^E$ | kJ/mol | 631 | 2 | 34 | 35 | T | ✓ | ✓ | ✓ |

**DiffMix (3 tasks)** Battery electrolytes—mixtures of salts and solvents—have been optimized to facilitate ion transport, prevent electron transfer, and stabilize electrode-electrolyte interfaces to produce energy-dense and durable battery systems [72, 21]. The DiffMix dataset is a collection of three tasks centered around thermodynamic and transport properties predictions of electrolytes originally gathered by Zhu et al. [81]. This data is under the CC BY-NC-ND 4.0 license, and we therefore cannot include it as part of our dataset.

- **Excess molar enthalpy** $H_m^E$: The excess molar enthalpy reflects changes in intermolecular interactions that occur during the mixing of different components [77]. It shows the non-ideality of the final solution and gives an explanation about enthalpic effects [49]. In particular, differences in molecular shape, size, and interaction types between components—along with variations in temperature and pressure—can lead to either an increase or a decrease in excess molar enthalpy [36, 63]. DiffMix dataset includes 631 $H_m^E$ data points curated from literature, covering 34 unique mixtures composed of 35 organic compounds across varying compositions. We rescaled the original range of the DiffMix excess molar enthalpy task from J/mol to kJ/mol to avoid passing big values to the neural networks.

- **Excess molar volume** $V_m^E$: The excess molar volume represents the deviation from ideal mixing volume. It exhibits a non-linear dependence on mole fraction [10] and temperature [66]—often showing a U-shaped trend with concentration and a decrease in absolute values as temperature increases. At higher temperatures, the dependence may shift to an S-shaped profile, making accurate prediction particularly challenging [71]. DiffMix dataset includes 1069 $V_m^E$ data points curated from literature, covering 28 unique mixtures composed of 25 organic compounds.

- **Ionic conductivity** $\kappa$: The ionic conductivity of the electrolyte is known as a key parameter to evaluate the performance of the solution in practical engineering applications. In the context of batteries, $\kappa$ changes considerably with the change of the electrolyte concentration [78]. DiffMix dataset includes 24,822 mixtures of single-salt-ternary-solvent electrolyte solutions generated using Advanced Electrolyte Model [22], and covering arbitrary combinations of two unique salts and six organic carbonate solvents at different concentration.

Table 12: **Model performances across CHEMIXHUB tasks** on 5-fold random CV splits. The mean and standard deviation are reported.

(a) MAE ($\downarrow$)

| Molecular rep. | Mixture rep. | DiffMix | | |
|---|---|---|---|---|
| | | $\kappa$ | $V_m^E$ | $H_m^E$ |
| GNN | Attention | 0.205 ± 0.061 | 0.060 ± 0.004 | 0.029 ± 0.006 |
| | Deepsets | 0.306 ± 0.054 | 0.072 ± 0.004 | 0.062 ± 0.014 |
| MolT5 | XGB | 0.059 ± 0.002 | **0.042 ± 0.007** | 0.042 ± 0.004 |
| | Attention | 0.167 ± 0.164 | 0.056 ± 0.005 | 0.023 ± 0.003 |
| | Deepsets | **0.046 ± 0.006** | 0.062 ± 0.005 | **0.021 ± 0.002** |
| RDKit | XGB | 0.050 ± 0.001 | 0.045 ± 0.00 | 0.045 ± 0.006 |
| | Attention | 0.168 ± 0.064 | 0.079 ± 0.008 | 0.251 ± 0.123 |
| | Deepsets | 0.110 ± 0.011 | 0.074 ± 0.005 | 0.090 ± 0.065 |

(b) Pearson $\rho$ ($\uparrow$)

| Molecular rep. | Mixture rep. | DiffMix | | |
|---|---|---|---|---|
| | | $\kappa$ | $V_m^E$ | $H_m^E$ |
| GNN | Attention | 0.993 ± 0.004 | **0.950 ± 0.005** | 0.996 ± 0.004 |
| | Deepsets | 0.984 ± 0.007 | 0.946 ± 0.007 | 0.982 ± 0.006 |
| MolT5 | XGB | 0.998 ± 0.000 | 0.933 ± 0.023 | 0.989 ± 0.003 |
| | Attention | 0.994 ± 0.010 | 0.950 ± 0.008 | 0.998 ± 0.001 |
| | Deepsets | **1.000 ± 0.000** | 0.949 ± 0.009 | **0.998 ± 0.000** |
| RDKit | XGB | 0.999 ± 0.000 | 0.932 ± 0.026 | 0.983 ± 0.010 |
| | Attention | 0.995 ± 0.003 | 0.944 ± 0.005 | 0.422 ± 0.378 |
| | Deepsets | 0.998 ± 0.000 | 0.945 ± 0.006 | 0.964 ± 0.052 |

(c) Kendall $\tau$ ($\uparrow$)

| Molecular rep. | Mixture rep. | DiffMix | | |
|---|---|---|---|---|
| | | $\kappa$ | $V_m^E$ | $H_m^E$ |
| GNN | Attention | 0.929 ± 0.019 | 0.873 ± 0.023 | 0.928 ± 0.028 |
| | Deepsets | 0.887 ± 0.013 | 0.863 ± 0.026 | 0.852 ± 0.031 |
| MolT5 | XGB | 0.973 ± 0.002 | **0.901 ± 0.025** | 0.909 ± 0.026 |
| | Attention | 0.948 ± 0.039 | 0.890 ± 0.022 | 0.957 ± 0.005 |
| | Deepsets | **0.983 ± 0.002** | 0.881 ± 0.023 | **0.957 ± 0.002** |
| RDKit | XGB | 0.980 ± 0.001 | 0.900 ± 0.025 | 0.903 ± 0.012 |
| | Attention | 0.945 ± 0.015 | 0.838 ± 0.045 | 0.472 ± 0.257 |
| | Deepsets | 0.954 ± 0.006 | 0.850 ± 0.027 | 0.828 ± 0.098 |

## A.12 Modeling salts

Salts are often present in mixtures, these are non-bonded small molecules that are found in the same environment as the molecule. To explore how to properly model salts we first look at if they contribute meaningfully to basic featurizations.

We constructed a 200-dimensional molecular embedding space using RDKIT 2D descriptors obtained from DESCRIPTASORUS [29], incorporating both salts and fragments for all the tasks in CHEMIXHUB. The number of unique salts is 824, and the number of fragments is 476. This space was projected into two dimensions using UMAP to visualize structural relationships, Figure 5. As shown in the UMAP plot, The resulting plot shows that salts (blue triangles) and fragments (orange circles) broadly co-localize, with many salts embedded near fragment clusters. To quantify these observations, we computed cosine distances between each salt and the fragment-only descriptor space. The resulting

distribution confirms that the vast majority of salts lie within a narrow cosine distance range centered around 0.04–0.05, with very few exceeding 0.1, Figure 6. In RDKIT descriptor space, such low distances imply near-identity in structural features. From these, we can observe that most salts appear to retain descriptor-level similarity to their constituent fragments. However, there is a subset which introduces structural changes significant enough to shift them away from the fragment space.

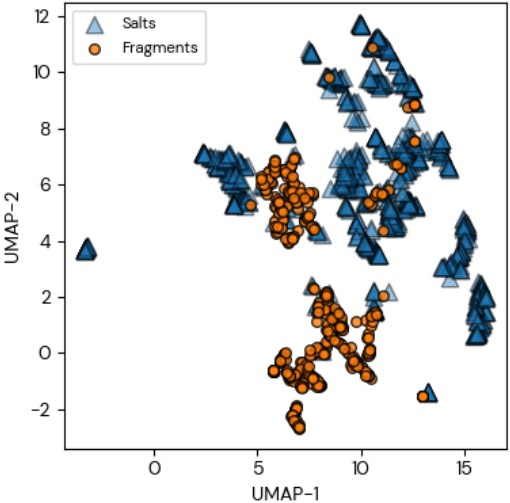

Figure 5: **The embedding space of salts and fragments in CHEMIXHUB**. UMAP projection of the combined RDKIT 2D descriptor space (200 dimensions) for salts and fragments. The embedding reveals well-defined structural clusters with apparent separation between salts and fragments, rather than overlap. Most salts appear in peripheral regions relative to the fragment clusters, suggesting distinct structural patterns at the descriptor level.

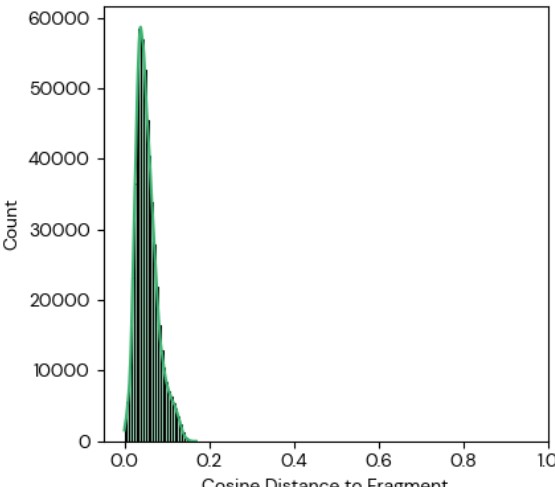

Figure 6: **Distribution of cosine distances**. The majority of salts fall within a tight cosine distance range (centered around 0.04–0.05), indicating strong structural similarity at the descriptor level. A smaller subset of salts shows higher distances, suggesting meaningful deviations from fragment-like chemistry.

Based on this analysis we conclude that most basic featurizations do not properly model salts. We think the best way to currently model salts is either as disconnected nodes in a graph. When using a GRAPHNETS architecture, these disconnected nodes get routed to the globals, so they are roughly equivalent to learnable salt-specific embeddings at the globals level of the graph.

