| $0.948 \pm 0.076$ | $\mathbf{0.974 \pm 0.003}$ | $0.998 \pm 0.000$ | $0.993 \pm 0.001$ | $0.970 \pm 0.004$ | $0.980 \pm 0.002$ |
| | Deepsets | $\mathbf{0.999 \pm 0.000}$ | $\underline{0.974 \pm 0.004}$ | $0.851 \pm 0.296$ | $0.996 \pm 0.001$ | $0.969 \pm 0.010$ | $0.981 \pm 0.002$ |
| MolT5 | XGB | $0.992 \pm 0.001$ | $0.924 \pm 0.005$ | $0.987 \pm 0.001$ | $\mathbf{0.999 \pm 0.000}$ | $\underline{0.976 \pm 0.001}$ | $0.989 \pm 0.001$ |
| | Attention | $0.998 \pm 0.001$ | $0.976 \pm 0.003$ | $0.997 \pm 0.004$ | $0.992 \pm 0.005$ | $0.973 \pm 0.001$ | $0.975 \pm 0.038$ |
| | Deepsets | $0.997 \pm 0.001$ | $0.976 \pm 0.003$ | $\mathbf{0.999 \pm 0.000}$ | $0.983 \pm 0.003$ | $0.967 \pm 0.002$ | $0.977 \pm 0.002$ |
| RDKit | XGB | $0.992 \pm 0.000$ | $0.945 \pm 0.007$ | $0.986 \pm 0.001$ | $\mathbf{0.999 \pm 0.000}$ | $\mathbf{0.977 \pm 0.001}$ | $\mathbf{0.992 \pm 0.000}$ |
| | Attention | $0.997 \pm 0.001$ | $0.972 \pm 0.003$ | $0.991 \pm 0.003$ | $0.996 \pm 0.001$ | $0.947 \pm 0.008$ | $0.995 \pm 0.000$ |
| | Deepsets | $0.996 \pm 0.001$ | $0.954 \pm 0.005$ | $\mathbf{0.999 \pm 0.000}$ | $0.973 \pm 0.004$ | $0.963 \pm 0.003$ | $0.970 \pm 0.002$ |

| Molecular rep. | Mixture rep. | IlThermo | | MON | NIST | Olfaction |
|---|---|---|---|---|---|---|
| | | $\ln(\kappa)$ | $\ln(\eta)$ | MON | $\ln(\eta)$ | Mixture similarity |
| GNN | Attention | $0.986 \pm 0.008$ | $0.975 \pm 0.012$ | $0.360 \pm 0.300$ | $\underline{0.990 \pm 0.002}$ | $0.447 \pm 0.120$ |
| | Deepsets | $0.989 \pm 0.002$ | $0.939 \pm 0.060$ | $0.820 \pm 0.087$ | $\underline{0.990 \pm 0.002}$ | $0.132 \pm 0.103$ |
| MolT5 | XGB | $\mathbf{0.997 \pm 0.000}$ | $\mathbf{0.995 \pm 0.001}$ | $0.860 \pm 0.032$ | $0.950 \pm 0.003$ | $0.432 \pm 0.047$ |
| | Attention | $0.988 \pm 0.001$ | $0.993 \pm 0.003$ | $0.893 \pm 0.028$ | $\mathbf{0.991 \pm 0.001}$ | $\mathbf{0.559 \pm 0.040}$ |
| | Deepsets | $0.993 \pm 0.002$ | $0.986 \pm 0.002$ | $0.880 \pm 0.036$ | $0.981 \pm 0.001$ | $0.548 \pm 0.025$ |
| RDKit | XGB | $\mathbf{0.997 \pm 0.001}$ | $\mathbf{0.995 \pm 0.001}$ | $\mathbf{0.913 \pm 0.019}$ | $0.957 \pm 0.003$ | $0.476 \pm 0.062$ |
| | Attention | $0.987 \pm 0.003$ | $0.981 \pm 0.003$ | $0.197 \pm 0.351$ | $0.977 \pm 0.024$ | $0.056 \pm 0.130$ |
| | Deepsets | $0.991 \pm 0.002$ | $0.992 \pm 0.001$ | $0.752 \pm 0.155$ | $0.980 \pm 0.002$ | $-0.091 \pm 0.050$ |

Table 9: **Model performances across CHEMIXHUB tasks** Reported Kendall ranking coefficient $\tau$ ($\uparrow$) on 5-fold random CV splits. The mean and standard deviation are reported.

| Molecular rep. | Mixture rep. | Miscible Solvents | | | Drug Solubility | SPE | NIST-full |
|---|---|---|---|---|---|---|---|
| | | $\rho$ | $\Delta H_{\mathrm{mix}}$ | $\Delta H_{\mathrm{vap}}$ | $\ln(S)$ | $\ln(\kappa)$ | $\ln(\eta)$ |
| GNN | Attention | $0.910 \pm 0.091$ | $\underline{0.835 \pm 0.004}$ | $0.969 \pm 0.002$ | $0.932 \pm 0.003$ | $0.868 \pm 0.016$ | $0.904 \pm 0.007$ |
| | Deepsets | $\mathbf{0.973 \pm 0.000}$ | $0.833 \pm 0.003$ | $0.816 \pm 0.318$ | $0.949 \pm 0.004$ | $0.869 \pm 0.023$ | $0.905 \pm 0.002$ |
| MolT5 | XGB | $0.924 \pm 0.00$ | $0.730 \pm 0.008$ | $0.897 \pm 0.003$ | $\mathbf{0.978 \pm 0.001}$ | $\mathbf{0.899 \pm 0.003}$ | $0.950 \pm 0.000$ |
| | Attention | $0.963 \pm 0.006$ | $\mathbf{0.835 \pm 0.002}$ | $0.955 \pm 0.034$ | $0.935 \pm 0.022$ | $0.881 \pm 0.003$ | $0.956 \pm 0.003$ |
| | Deepsets | $0.966 \pm 0.002$ | $\mathbf{0.835 \pm 0.002}$ | $\mathbf{0.976 \pm 0.001}$ | $0.893 \pm 0.010$ | $0.861 \pm 0.004$ | $0.910 \pm 0.004$ |
| RDKit | XGB | $0.929 \pm 0.002$ | $0.773 \pm 0.005$ | $0.898 \pm 0.003$ | $\mathbf{0.978 \pm 0.001}$ | $\underline{0.899 \pm 0.004}$ | $\mathbf{0.966 \pm 0.000}$ |
| | Attention | $0.961 \pm 0.003$ | $0.829 \pm 0.003$ | $0.944 \pm 0.012$ | $0.948 \pm 0.006$ | $0.840 \pm 0.014$ | $0.957 \pm 0.003$ |
| | Deepsets | $0.956 \pm 0.001$ | $0.788 \pm 0.008$ | $0.973 \pm 0.002$ | $0.856 \pm 0.009$ | $0.855 \pm 0.007$ | $0.921 \pm 0.002$ |

| Molecular rep. | Mixture rep. | IlThermo | | MON | NIST | Olfaction |
|---|---|---|---|---|---|---|
| | | $\ln(\kappa)$ | $\ln(\eta)$ | MON | $\ln(\eta)$ | Mixture similarity |
| GNN | Attention | $0.923 \pm 0.021$ | $0.941 \pm 0.029$ | $0.348 \pm 0.203$ | $0.940 \pm 0.007$ | $0.312 \pm 0.073$ |
| | Deepsets | $0.930 \pm 0.006$ | $0.863 \pm 0.103$ | $0.687 \pm 0.093$ | $\mathbf{0.942 \pm 0.009}$ | $0.166 \pm 0.067$ |
| MolT5 | XGB | $\mathbf{0.974 \pm 0.000}$ | $\underline{0.967 \pm 0.001}$ | $0.756 \pm 0.038$ | $0.863 \pm 0.002$ | $0.319 \pm 0.047$ |
| | Attention | $0.930 \pm 0.004$ | $\underline{0.967 \pm 0.010}$ | $0.768 \pm 0.033$ | $0.939 \pm 0.001$ | $0.377 \pm 0.042$ |
| | Deepsets | $0.942 \pm 0.004$ | $0.945 \pm 0.002$ | $0.714 \pm 0.012$ | $0.896 \pm 0.001$ | $\mathbf{0.390 \pm 0.011}$ |
| RDKit | XGB | $\underline{0.973 \pm 0.001}$ | $\mathbf{0.968 \pm 0.001}$ | $\mathbf{0.781 \pm 0.029}$ | $0.883 \pm 0.003$ | $0.342 \pm 0.040$ |
| | Attention | $0.924 \pm 0.015$ | $0.957 \pm 0.000$ | $0.164 \pm 0.266$ | $0.916 \pm 0.029$ | $0.036 \pm 0.065$ |
| | Deepsets | $0.928 \pm 0.005$ | $0.956 \pm 0.002$ | $0.583 \pm 0.121$ | $0.897 \pm 0.005$ | $-0.048 \pm 0.035$ |