# OpenReview forum: "CheMixHub: Datasets and Benchmarks for Chemical Mixture Property Prediction"
_NeurIPS.cc/2025/Datasets_and_Benchmarks_Track — NeurIPS 2025 Datasets and Benchmarks Track poster_

### Official Review · Reviewer_VtoY · 2025-06-30

**Rating:** 3
**Confidence:** 3

**Summary:**

This paper introduces CheMixHub, a comprehensive benchmark for predicting properties of chemical mixtures, addressing a gap in machine learning for multi-component systems. The paper curates 11 tasks from 7 publicly available datasets, totaling ~500k data points and covers diverse applications.

**Dataset Code Accessibility:**

Yes

**Ethical Considerations:**

No, there are no or only very minor ethics concerns

**Final Justification:**

The practical significance of this work remains unclear to me. I am inclined to maintain the current borderline score.

**Limitations Weaknesses:**

1. Although the paper proposes a unified benchmark, it does not study whether the knowledge from one task can be transferred to other tasks.
2. The tasks of chemical mixture prediction appear in various applications. It may be not easy to organize all the tasks in one benchmark. The models or features that perform well may be very task-dependent, where domain expertise is critical.
3. In real applications, the scenario of few-shot or zero-shot learning is important. It would be better if the benchmark could cover these.

**Strengths Contributions:**

1. The task of chemical mixture property prediction is ubiquitous in industry but has not been explored very much in ML.
2. The design of the deep learning modeling space for chemical mixtures is clear and grounded in prior works.
3. The paper is generally well-written and easy to follow.

---

> ### Author Rebuttal · Authors · 2025-07-31
>
> We appreciate the reviewer for their feedback on our work.
>
> > Although the paper proposes a unified benchmark, it does not study whether the knowledge from one task can be transferred to other tasks.
>
> We thank the reviewer for the insightful comment and agree that investigating transfer learning would greatly enrich our modeling landscape overview. We performed additional experiments to cover transfer learning, focusing on answering the following research questions:
>
> > 1. If we have one dataset of mixtures for which several properties have been measured and separate models are trained on each property, how does this model perform when fine-tuned on other tasks?
>
> We evaluated transfer learning capabilities of models trained on the Miscible Solvent dataset, which contains 3 property prediction tasks (Density $\rho$, $\Delta H_{mix}$ and $\Delta H_{vap}$) spanning the same set of chemical mixtures. We compare these fine-tuned models to the best performing models for these tasks found in Section 4.2 (see Table 1).
>
> We see that fine-tuning the best Deep Learning models for each task (according to Section 4.2) on other task can yield dramatic differences depending on the task the model was originally trained on: Models initially trained to predict highly non-linear properties — harder tasks — like $\Delta H_{mix}$ and $\Delta H_{vap}$ perform really well when finetuned to predict density $\rho$ but the model initially trained to predict density $\rho$ fails at delivering good performances on $\Delta H_{mix}$ and $\Delta H_{vap}$ predictions. Architectural differences may also play a role in this phenomenon.
>
> **Table 1. Transfer learning capabilities of models within the Miscible Solvent dataset tasks.** Metrics are reported on 5-fold random CV splits. The mean and standard deviation are reported. The best model statistics are taken from Section 4.2 and Appendix A.8. FT is the acronym for "finetuned", MS is the acronym for "miscible solvent".
> |Model|Pearson $\rho$ ($\uparrow$) |MAE ($\downarrow$) | Kendall $\tau$ ($\uparrow$)|
> |-|-|-|-|
> |||*MS - Density $\rho$ results*||
> |$\rho$ Best model in Section 4.2|**0.999 ± 0.000**|**0.003 ± 0.000**|**0.973 ± 0.000**|
> |$\Delta H_{mix}$-FT|0.955 ± 0.006|0.021 ± 0.001|0.824 ± 0.009|
> |$\Delta H_{vap}$-FT|0.929 ± 0.008|0.026 ± 0.002|0.769 ± 0.018|
> |||*MS - $\Delta H_{vap}$ results*||
> |$\Delta H_{vap}$ Best model in Section 4.2|**0.999 ± 0.000**|**0.071 ± 0.002**|**0.976 ± 0.001**|
> |$\Delta H_{mix}$-FT|0.808 ± 0.017|1.063 ± 0.057|0.611 ± 0.034|
> |$\rho$-FT|0.644 ± 0.088|1.366 ± 0.176|0.465 ± 0.074|
> |||*MS - $\Delta H_{mix}$ results*||
> |$\Delta H_{mix}$ Best model in Section 4.2|**0.976 ± 0.003**|**0.157 ± 0.002**|**0.835 ± 0.002**|
> |$\Delta H_{vap}$-FT|0.626 ± 0.022|0.527 ± 0.008|0.439 ± 0.025|
> |$\rho$-FT|0.348 ± 0.044|0.629 ± 0.013|0.237 ± 0.033|
>
> > 2. If we have two datasets of mixtures probing very different properties,  how does their respective best-performing model do when fine-tuned on the other task?
>
> We evaluated transfer learning capabilities of two models trained on different datasets and tasks: the best deep learning model trained on the Miscible Solvent $\Delta H_{vap}$ task and the other one trained on the Motor Octane Number (MON) task (according to Section 4.2). We compare these fine-tuned models to the best performing models for these tasks found in Section 4.2 (see Table 2).
>
> We observe a simple fine-tuning approach of the best Deep Learning models for each task on another task from a different dataset does not yield good performance, especially compared to "in-dataset" finetuning results above, which could suggest the models are overfitting to their respective tasks. An interesting experimental set up to further answer this questions would be to evaluate multi-task learning capabilities of these models across datasets, which should be easily implementable thanks to our unified framework.
>
> **Table 2. Transfer learning capabilities of models across the Miscible Solvent $\Delta H_{vap}$ task and the Motor Octane Number (MON) task.** Metrics are reported on 5-fold random CV splits. The mean and standard deviation are reported. The best model statistics are taken from Section 4.2 and Appendix A.8. FT is the acronym for "finetuned", MS is the acronym for "miscible solvent".
> |Model|Pearson $\rho$ ($\uparrow$) |MAE ($\downarrow$) | Kendall $\tau$ ($\uparrow$)|
> |-|-|-|-|
> |||*MON*||
> |MON Best model in Section 4.2|0.913 ± 0.019|4.570 ± 0.348|0.781 ± 0.029|
> |$\Delta H_{vap}$-FT|0.160 ± 0.108 | 33.199 ± 1.606 | 0.144 ± 0.056|
> |||*MS - $\Delta H_{vap}$*||
> |$\Delta H_{vap}$ Best model in Section 4.2|0.999 ± 0.000|0.071 ± 0.002|0.976 ± 0.001|
> |MON-FT|0.501 ± 0.095|1.582 ± 0.095|0.296 ± 0.067|
>
>
> The results presented above will be included in the camera-ready version of CheMixHub and the code for these experiments can be found on Github under `scripts/run_transfer.py`.
>
> ---
>
> > The tasks of chemical mixture prediction appear in various applications. It may be not easy to organize all the tasks in one benchmark. The models or features that perform well may be very task-dependent, where domain expertise is critical.
>
> We agree with the reviewer on the fact that chemical mixtures span multiple application domains and that it is not easy to organize all the tasks within one benchmark. However, we firmly believe that standardizing these tasks and evaluating models on each of them according to the same standards is crucial to establish solid baselines and best practices for future model development. Our approach ressembles MoleculeNet, which gathers many different tasks for single molecule property prediciton spanning many applications and scales and catalyzed model development [1]. Moreover, having these wide range of standardized tasks within the same framework lowers the barrier to the development of foundational models for chemical mixtures.
>
> [1] Wu, Zhenqin, et al. "MoleculeNet: a benchmark for molecular machine learning." Chemical science 9.2 (2018): 513-530.
>
> ---
>
> > In real applications, the scenario of few-shot or zero-shot learning is important. It would be better if the benchmark could cover these.
>
> We thank the reviewer for their suggestion. We performed additional experiments to investigate the zero-shot capabilities of the best performing models for each of the Viscosity ($\eta$) prediction tasks in CheMixHub and observe good zero shot capabilities for tasks that have similar viscosity value ranges (see Table 3).
>
> **Table 3. Zero shot learning capabilities of models across the Viscosity ($\eta$) prediction tasks in CheMixHub.** Results for 5-fold random CV splits. The mean and standard deviation are reported. The best model statistics are taken from Section 4.2 and Appendix A.8. FT is the acronym for "finetuned", MS is the acronym for "miscible solvent"
> |Model|Pearson $\rho$ ($\uparrow$) |MAE ($\downarrow$) | Kendall $\tau$ ($\uparrow$)|
> |-|-|-|-|
> |||*NIST*||
> |Best model NIST|**0.991 ± 0.001**|**0.030 ± 0.001**|**0.939 ± 0.001**|
> |NIST-full-FT|0.985 ± 0.002|6.806 ± 0.012|0.926 ± 0.004|
> |IlThermo Viscosity-FT|0.575 ± 0.028|5.880 ± 0.129|0.451 ± 0.024|
> |||*NIST-full*||
> |Best model NIST-full|**0.992 ± 0.000**|**0.055 ± 0.000**|0.966 ± 0.000|
> |IlThermo Viscosity-FT|0.775 ± 0.018|0.811 ± 0.078|**1.000 ± 0.000**|
> |NIST-FT|0.694 ± 0.021|6.281 ± 0.005|**1.000 ± 0.000**|
> |||*IlThermo Viscosity*||
> |Best model|**0.995 ± 0.001**|**0.076 ± 0.002**|**0.968 ± 0.001**|
> |NIST-full-FT|0.956 ± 0.004|0.276 ± 0.032|0.880 ± 0.015|
> |NIST-FT|0.452 ± 0.041|4.815 ± 0.030|0.330 ± 0.047|
>
> The results presented above will be included in the camera-ready version of CheMixHub and the code for these experiments can be found on Github under `scripts/run_zero_shot.py`.
>
> ---
>
> We hope that the additional experiments we have conducted after the reviewer feedback bring to the reviewer a new perspective on this work and help them appreciate it more. If so, we would appreciate it if the reviewer could raise their score.

---

> > ### Comment · Reviewer_VtoY · 2025-08-07
> >
> > Thank you for the rebuttal. I am inclined to maintain the current borderline score.

---

> > > ### Author Response · Authors · 2025-08-08
> > >
> > > We are sorry that our new experiments and discussions did not satisfy the reviewer. We are happy to receive feedback  on the additional experiments conducted and answer any remaining questions the reviewer might have on our work. We thank the reviewer for their time.

---

### Official Review · Reviewer_uojg · 2025-07-01

**Rating:** 5
**Confidence:** 3

**Summary:**

This paper introduces CheMixHub, a well-designed benchmark for machine learning on chemical mixtures, a highly relevant but understudied area. The benchmark spans 11 property prediction tasks across domains such as drug delivery and battery electrolytes, using data from 7 public datasets (~500k examples). A key strength is the inclusion of diverse and realistic data splits, which support robust evaluation of generalization.

The authors also provide an extensive set of baseline evaluations across classical models, GNNs, and chemical language models (CLMs), revealing important insights, e.g., about the CLM model when combined with physics-based constraints. The dataset and code are publicly released, making this a valuable resource for the community. Overall, this is a timely and impactful benchmark that fills an important gap and lays a strong foundation for future research in mixture modeling.

**Dataset Code Accessibility:**

Yes

**Dataset Code Comments:**

Detailed for Code and Datasets are provided in GitHub.

**Ethical Considerations:**

No, there are no or only very minor ethics concerns

**Final Justification:**

I would keep my current positive scores. In general, I strongly support to accept this paper.

**Limitations Weaknesses:**

The reviewer did not identify any major limitations in the benchmark design or experimental framework. However, one potential area for extension concerns the graph representations used in the baseline models. Recent studies [1, 2] have demonstrated the benefits of integrating 2D molecular graphs with 3D conformer information for molecular property prediction. It would be valuable to explore whether such structure-aware approaches could improve performance in the chemical mixture tasks presented in CheMixHub. The authors can consider adding an experiment—perhaps on a selected subset of tasks—to assess the potential gains from incorporating 3D conformers.


[1] Nguyen, Duy MH, et al. "Structure-aware E(3)-Invariant Molecular Conformer Aggregation Networks." ICML 2024.

[2] Zhu, Yanqiao, et al. "Learning Over Molecular Conformer Ensembles: Datasets and Benchmarks." ICLR 2024.

**Strengths Contributions:**

Reviewer found the following major contributions and strengths of the paper:

**Novelty in Benchmark Design**
- The paper addresses an important and underexplored problem by introducing the first standardized benchmark for chemical mixtures, which fills a clear gap in the molecular machine learning literature.

- The benchmark is thoughtfully designed, covering a diverse set of real-world chemical mixture prediction tasks, and includes multiple challenging data splitting strategies that go beyond standard random splits. These enable more realistic evaluations of generalization.

- The authors provide a well-structured modeling framework, with strong baselines spanning multiple representation types (GNNs, CLMs, RDKit descriptors) and mixture aggregation strategies, offering a valuable starting point for future work.

- A key contribution is the demonstration that incorporating physics-based priors, such as the Arrhenius equation, significantly improves performance on out-of-distribution splits and enhances interpretability.

- The benchmark is fully open-sourced, with detailed documentation and reproducibility protocols, making it accessible and useful to the broader research community.

**Writing**

The paper is well-written and easy to follow with good demonstrated figures.

---

> ### Author Rebuttal · Authors · 2025-07-31
>
> We thank the reviewer for their insightful comments and suggestions on our work, and appreciate that the reviewer finds our work useful and impactful.
>
> > [..] one potential area for extension concerns the graph representations used in the baseline models. Recent studies have demonstrated the benefits of integrating 2D molecular graphs with 3D conformer information for molecular property prediction. It would be valuable to explore whether such structure-aware approaches could improve performance in the chemical mixture tasks presented in CheMixHub. The authors can consider adding an experiment—perhaps on a selected subset of tasks—to assess the potential gains from incorporating 3D conformers.
>
> We appreciate the reviewer’s insightful suggestion regarding the integration of 3D conformational information to enhance molecular property prediction. Indeed, recent studies have demonstrated the advantages of combining molecular graphs with 3D conformers in various contexts. However, molecular flexibility is a critical factor in determining the effectiveness of such approaches. Notably, Zhu et al. (ICLR 2024) [1] emphasized that the benefits of conformer ensemble modeling are most pronounced for highly flexible molecules—specifically, those possessing a substantial number of rotatable bonds. In their benchmark study involving a wide variety of molecules, Zhu et al. focused exclusively on compounds with more than five rotatable bonds, reporting an average of over 6.99 rotatable bonds per molecule across all datasets. Molecules with fewer rotatable bonds were explicitly excluded to ensure that the variability in 3D conformers would meaningfully impact molecular properties.
>
> In contrast, the CheMixHub benchmark emphasizes chemically diverse mixtures —including ionic liquids, organic solvents, and electrolyte components — many of which are structurally rigid. Our analysis indicates that over 76.4% of the molecules in our dataset possess fewer than five rotatable bonds, with 50% having fewer than two. For such relatively rigid molecules, the number of low-energy conformers is limited, and the resulting structures are often quite similar. In these cases, a single representative pose or even a molecular graph encoding is sufficient to capture the relevant shape-determining information. Consequently, generating an ensemble of conformers may introduce unnecessary computational overhead and noise without a corresponding gain in predictive performance.
>
> A more systematic investigation of conformer ensemble modeling for mixtures in the presence of highly flexible molecules would be an interesting topic for future work and could form the basis of a separate study.
>
> [1] Zhu, Y., Hwang, J., Adams, K., Liu, Z., Nan, B., Stenfors, B., ... & Wang, W. (2023). Learning over molecular conformer ensembles: Datasets and benchmarks. arXiv preprint arXiv:2310.00115.

---

> > ### Comment · Reviewer_uojg · 2025-08-09
> > **Response to Authors**
> >
> > Thank you, Authors, for your clarification. It would be great if you could include these things in the discussion of the paper.
> > Regarding rating, I would keep my original positive review.

---

### Official Review · Reviewer_MoLL · 2025-07-06

**Rating:** 5
**Confidence:** 4

**Summary:**

Nearly all chemical products are composed of mixtures of multiple compounds, making the prediction of chemical mixture properties critically important. However, existing machine learning research has primarily focused on single-molecule systems, with relatively limited attention given to multi-component mixtures. A key reason for the slow progress in this area is the lack of standardized datasets. This article introduces CheMixHub, a dataset and benchmarking platform specifically designed for the prediction of chemical mixture properties.

**Additional Feedback:**

1. Although CheMixHub already covers a variety of tasks such as drug solubility, electrolyte conductivity, and olfaction, it is recommended that future work include more challenging or practically relevant scenarios, such as: complex emulsion systems (e.g., in food or cosmetics); battery electrolyte formulation tasks balancing performance and safety; and high-component (>5) fuel or lubricant formulation data.
2. The authors implemented different modeling hierarchies using basic modules such as DeepSets and Self-Attention, and presented a systematic model space partitioning (Figure 3), which is a major highlight of the paper. Future work could explore more expressive architectures, such as:
    - Graph of Graphs (GoG): for nested modeling of complex molecular sets
    - Hypergraph Transformers: to capture high-order interactions among components
    - Causal GNNs/Attention mechanisms: to enhance interpretability under formulation changes
3. Adding the data source (e.g., NIST/ILThermo) for each task in Table 1 would improve clarity. In Section 4 “Benchmark”, it is suggested to include error curves or performance distribution plots for representative tasks to strengthen the comparative analysis.

**Dataset Code Accessibility:**

Yes

**Dataset Code Comments:**

The authors have provided complete code for dataset processing and model training for CheMixHub, and they demonstrate good design for accessibility and reproducibility in multiple aspects.

**Ethical Comments:**

No, this study does not raise major ethical concerns.

The research focuses solely on the construction of datasets and benchmarks for predicting the properties of chemical mixtures. All data are sourced from publicly available literature and databases (such as ILThermo and NIST). It does not involve human or animal experiments, nor does it develop generative models for material or drug design. Moreover, the authors explicitly state, *“We condemn any malicious use of our work for the synthesis of hazardous chemicals”* (end of Section 5, line 316), indicating that there are no apparent ethical risks or negative societal impacts.

**Ethical Considerations:**

No, there are no or only very minor ethics concerns

**Final Justification:**

I recommend an Accept (5) as the authors provided a satisfactory rebuttal that adequately addressed the initial concerns.

**Limitations Weaknesses:**

1. All tasks in the paper are formulated as forward prediction—i.e., predicting properties from component structures and conditions (Section 4). No task reflects the ability to infer mixture components from target properties, which limits the model's applicability in formulation optimization and materials discovery. The task setup could be extended to include inverse modeling or conditional generative models (e.g., CVAE), allowing the design of candidate formulations from desired property targets—such as searching for mixtures with specified conductivity and low viscosity.
2. The authors attempted to incorporate the Arrhenius equation as a physical constraint in the model (Section 4.4) to improve generalization across temperatures. However, this approach is only applicable to properties that follow this equation (e.g., viscosity, conductivity). Other properties, such as enthalpy of evaporation or mixing enthalpy, were not treated similarly. Future work could integrate additional known physical relationships (e.g., Flory–Huggins, NRTL, SAFT models), and explore approaches like symbolic regression or physics-informed graph neural networks (PGNNs) to learn complex or non-differentiable physical functions.
3. Although the paper implements various deep learning architectures (Figure 3), it does not provide any explanation for how the models make predictions—for instance, which components or molecular features are most influential. Interpretability could be enhanced by incorporating graph attention mechanisms (e.g., GAT) or post hoc interpretation methods (e.g., SHAP, Integrated Gradients) to analyze model behavior, improving transparency and practical utility.
4. While the authors constructed various generalization splits (Section 3.4, Figure 4), all evaluations were conducted within the dataset, without closed-loop validation against experimental data or molecular simulation tools. To demonstrate the real-world applicability of the predictions, high-confidence formulations could be selected for literature comparison or simulation-based validation (e.g., molecular dynamics, COSMO-RS).

**Strengths Contributions:**

1. The CheMixHub dataset includes 11 tasks (such as drug solubility, electrolyte conductivity, viscosity, olfactory similarity, octane number, etc.) collected from 7 public data sources, totaling approximately 500,000 data points. It covers mixtures ranging from binary to complex multi-component systems.
2. The authors constructed two new tasks—viscosity and ionic conductivity prediction—based on the ILThermo database.
3. The dataset offers four different splitting strategies (random split, unseen components, varying mixture size/complexity, and out-of-distribution temperature data) to evaluate model generalization in diverse scenarios.
4. The study evaluates a variety of methods, including graph neural networks (GNNs), chemical language models (CLMs) such as MolT5, and traditional descriptors (e.g., RDKit). It compares the performance of traditional machine learning models like XGBoost with deep learning approaches.
5. The authors incorporate a physics-based model derived from the Arrhenius equation (for temperature-dependent viscosity and conductivity prediction), which significantly improves performance under more challenging data splits.

---

> ### Author Rebuttal · Authors · 2025-07-31
>
> We thank the reviewer for their insightful comments and suggestions on our work, and appreciate that the reviewer finds our work useful and impactful.
>
> > All tasks in the paper are formulated as forward prediction—i.e., predicting properties from component structures and conditions (Section 4). No task reflects the ability to infer mixture components from target properties, which limits the model's applicability in formulation optimization and materials discovery. The task setup could be extended to include inverse modeling or conditional generative models (e.g., CVAE), allowing the design of candidate formulations from desired property targets—such as searching for mixtures with specified conductivity and low viscosity.
>
> It would be possible to extend CheMixHub to generation or conditional generation tasks if predictors are used as oracle or surrogates. Simulation-based validation (eg. Molecular Dynamics) could also be used but would be much more expensive, as the reviewer rightly pointed out. Additional metrics regarding eg. novelty, synthetic accessibility of generated mixtures would also have to be implemented. Finally, generation would have to handle permutation invariance decoding which is possible. This is a promising line of work that we would want to tackle in the future and for which robust benchmarking efforts like CheMixHub constitute a stepping stone.
>
> ---
>
> > The authors attempted to incorporate the Arrhenius equation as a physical constraint in the model (Section 4.4) to improve generalization across temperatures. However, this approach is only applicable to properties that follow this equation (e.g., viscosity, conductivity). Other properties, such as enthalpy of evaporation or mixing enthalpy, were not treated similarly. Future work could integrate additional known physical relationships (e.g., Flory–Huggins, NRTL, SAFT models), and explore approaches like symbolic regression or physics-informed graph neural networks (PGNNs) to learn complex or non-differentiable physical functions.
>
> We thank the reviewer for this suggestion and agree and this would constitute a great future line of work.
>
> ---
>
> > Although the paper implements various deep learning architectures (Figure 3), it does not provide any explanation for how the models make predictions—for instance, which components or molecular features are most influential. Interpretability could be enhanced by incorporating graph attention mechanisms (e.g., GAT) or post hoc interpretation methods (e.g., SHAP, Integrated Gradients) to analyze model behavior, improving transparency and practical utility.
>
> We thank the reviewer for this suggestion and agree and this would constitute a great future line of work, which merits a dedicated article. We have hinted at this current limitation of machine learning models applied to the chemical mixture space, and have highlighted some works that have started opening up this research direction [1].
>
> [1] Tom, Gary, et al. "From Molecules to Mixtures: Learning Representations of Olfactory Mixture Similarity using Inductive Biases." arXiv preprint arXiv:2501.16271 (2025).
>
> ---
>
> > While the authors constructed various generalization splits (Section 3.4, Figure 4), all evaluations were conducted within the dataset, without closed-loop validation against experimental data or molecular simulation tools. To demonstrate the real-world applicability of the predictions, high-confidence formulations could be selected for literature comparison or simulation-based validation (e.g., molecular dynamics, COSMO-RS).
>
> We thank the reviewer for this comment, and believe this is of crucial importance to the field. CheMixHub did not consider running simulation-based validation as they are expensive and would require creating tailored protocols for each of the tasks. Finally, experimental validation is out of the scope of this paper, but we have highlighted some works that have started validating ML models by performing wet-lab experiments [1].
>
> [1] Ruza, Jurgis, et al. "Autonomous Discovery of Polymer Electrolyte Formulations with Warm-start Batch Bayesian Optimization." (2025).
>
> ---
>
> > Although CheMixHub already covers a variety of tasks such as drug solubility, electrolyte conductivity, and olfaction, it is recommended that future work include more challenging or practically relevant scenarios, such as: complex emulsion systems (e.g., in food or cosmetics); battery electrolyte formulation tasks balancing performance and safety; and high-component (>5) fuel or lubricant formulation data.
>
> We have done our best to gather and curate as much publicly available chemical mixture data, but the field remains dominated by industry and closed-source data gathering efforts. We hope that this work encourages the community to release more data to add to this standardization effort.
>
> Regarding complex emulsion systems, there exists some work that combine Bayesian Optimization with wet-lab experiments that have open sourced their data. However, including these datasets in our benchmark would require representing highly complex polymer structures, which remains an unsolved challenge in the field.
>
> [1] Chitre, Aniket, et al. "Accelerating formulation design via machine learning: generating a high-throughput shampoo formulations dataset." Scientific Data 11.1 (2024): 728.
>
> [2] Ros, Helena, et al. "Efficient discovery of new medicine formulations using a semi-self-driven robotic formulator." Digital Discovery (2025).
>
> ---
>
> > The authors implemented different modeling hierarchies using basic modules such as DeepSets and Self-Attention, and presented a systematic model space partitioning (Figure 3), which is a major highlight of the paper. Future work could explore more expressive architectures, such as: 1) Graph of Graphs (GoG): for nested modeling of complex molecular sets 2) Hypergraph Transformers: to capture high-order interactions among components 3) Causal GNNs/Attention mechanisms: to enhance interpretability under formulation changes
>
> We thank the reviewer for this suggestion and agree that benchmarking those architectures constitute an exciting line of work. The modeling modules provided in CheMixHub are meant as a basic baseline and we hope that the community will eagerly take part into testing these architectures and newer ones on the 11 tasks we have provided.
>
> ---
>
> > Adding the data source (e.g., NIST/ILThermo) for each task in Table 1 would improve clarity.
>
> We thank the reviewer for providing feedback on Table 1, we will add it to the camera-ready version of the paper. Additionally, we have added the data source on the GitHub `README.md`.
>
> ---
>
> > In Section 4 “Benchmark”, it is suggested to include error curves or performance distribution plots for representative tasks to strengthen the comparative analysis.
>
> We thank the reviewer for providing feedback on the Benchmark section. We have added performance distribution plots for each of the CheMixHub tasks on the GitHub under `figures/performances_plots.png` and we will add them to the camera-ready version of the paper.

---

### Official Review · Reviewer_QjXn · 2025-07-23

**Rating:** 5
**Confidence:** 4

**Summary:**

This paper introduces CheMixHub, a holistic benchmark for molecular mixtures. It encompasses approximately 500k data points curated from 7 publicly available datasets. The dataset is applied to 11 regression tasks, and a comprehensive benchmark is established for evaluation, contributing to the property prediction of multi-component compounds. The paper provides detailed descriptions of the dataset's motivation, data construction, splitting, application tasks, and benchmark.

**Additional Feedback:**

All my questions are written in "Limitations Weaknesses". If the authors can address my confusion, I will raise my score.

**Dataset Code Accessibility:**

Yes

**Dataset Code Comments:**

There is relatively detailed information about the models and hypeparameters in the appendix.

**Ethical Considerations:**

No, there are no or only very minor ethics concerns

**Final Justification:**

The author has addressed my confusion and questions. I have decided to increase the score.

**Limitations Weaknesses:**

The main limitations and questions are as follows:
1. The data description is not comprehensive enough. Could you provide detailed descriptive information of the data, such as the number of compositions, the average number of atoms per molecule, the maximum number of atoms, the minimum number of atoms, etc.?
2. Whether the properties of mixtures are related to the order. For example, different addition orders of solvent mixtures may affect their properties. In this case, is it reasonable to use permutation-invariant models for prediction? Are there cases in the dataset where the same mixture composition with different orders corresponds to different properties?
3. Are there multiple properties for one composition in the dataset, and how is this situation handled?
4. Can this dataset be extended to generation or conditional generation tasks besides property prediction? Can it be applied to molecular synthesis tasks, and what potential limitations may exist?
5. In lines 44 and 45, physics-based constraints vary across different application domains (such as pharmaceuticals, polymers, etc.). How are these constraints derived? And how is Equation 1 incorporated into the loss function?
6. Are the 3D conformations of molecules not considered in the benchmark? Is the 3D conformation also important for the prediction of interactions?

**Strengths Contributions:**

The paper is well-structured, with detailed descriptions from the introduction of the dataset, data processing, to the benchmark. Although the methods used lack innovation, the provision of such a dataset is valuable.
The main contributions are as follows:
1. Proposing a dataset that consolidates and standardizes 11 tasks from 7 datasets, reflecting the diversity and state of the chemical mixtures space, along with implementing four distinct data splitting methodologies .
2. Introducing 11 regressive tasks and establishing a comprehensive benchmark.

---

> ### Author Rebuttal · Authors · 2025-07-31
>
> We thank the reviewer for their insightful comments on our work. We address the limitations/questions from the review below:
>
> > The data description is not comprehensive enough. Could you provide detailed descriptive information of the data, such as the number of compositions, the average number of atoms per molecule, the maximum number of atoms, the minimum number of atoms, etc.?
>
> We have improved the description of our data by adding more statistics for each of the 11 tasks in CheMixHub. We have released the code to produce these statistics on our GitHub (under `scripts/run_group_features.py`) so users have access to a basic analysis they can deepen further depending on their need.
>
>
> **Table 1: Additional statistics on molecules for each of 11 tasks in CheMixHub**
>
> |Task|Avg # Atoms/Mol ± std|Max # Atoms/Mol|Min # Atoms/Mol|Avg # Fragments ± std|Max # Fragments|Avg MolWt ± std|Avg # Rotatable Bonds ± std| Avg Components Mixture ± std |
> |-|-|-|-|-|-|-|-|-|
> |Miscible solvents|8.28±3.17|18|3|1.0±0.0|1|123.73±43.96|3.40±3.17|3.72±1.08|
> |IlThermo Viscosity|15.80±9.28|77|1|1.76±0.54|4|250.91±145.56|5.12±6.13|2.21±0.41|
> |IlThermo Electrical conductivity|17.33±10.73|62|1|1.85±0.59|4|280.30±174.57|5.76±6.51|2.40±0.49|
> |NIST-full|12.90±8.98|95|1|1.50±0.70|8|203.98±135.28|4.00±5.60|1.88±0.33|
> |NIST|9.12±4.71|63|1|1.0±0.0|1|140.52±73.17|3.14±3.79|1.92±0.28|
> |Drug solubility|14.48±9.16|51|1|1.11±0.33|3|212.40±128.17|2.37±2.45|1.91±0.29|
> |Solid polymer electrolyte|30.86±47.75|676|2|1.24±0.44|3|473.36±738.30|18.11±33.19|2.24±0.67|
> |Olfactory similarity|9.53±3.43|21|3|1.0±0.0|1|135.67±45.03|2.72±2.29|13.30±10.51|
> |Motor Octane Number|7.93±1.94|12|2|1.0±0.0|1|110.66±26.19|1.71±1.69|5.69±14.24|
>
> ---
>
> > Whether the properties of mixtures are related to the order. For example, different addition orders of solvent mixtures may affect their properties. In this case, is it reasonable to use permutation-invariant models for prediction? Are there cases in the dataset where the same mixture composition with different orders corresponds to different properties?
>
> All chemical mixture properties are permutation-invariant — the order of the mixture components does not affect the properties — this is a unique symmetry of this problem space. In our camera-ready version we will make sure this is more clearly stated.
>
> ---
>
> > Are there multiple properties for one composition in the dataset, and how is this situation handled?
>
> Yes, in this case we treat them as separate tasks (cf. Miscible solvents). Future work could tackle multitask learning and general mixture representations.
>
> **We have conducted additional transfer learning experiments on the Miscible Solvents Dataset, where we investigate how a model trained on one of the tasks (eg. Density prediction) performs when finetuned on one of the other tasks (eg. Enthalpy of Mixing or Heat of Vaporization)**. Our results show that fine-tuning effectiveness is strongly influenced by the nature of the original training task: Training on harder tasks seems to help learning easier tasks when considering the same set of molecular mixtures, although architectural differences may influence this conclusion. Detailed results can be found in the response to reviewer VtoY.
>
> ---
>
> > Can this dataset be extended to generation or conditional generation tasks besides property prediction? Can it be applied to molecular synthesis tasks, and what potential limitations may exist?
>
> It would be possible to extend CheMixHub to generation or conditional generation tasks if predictors are used as oracle or surrogates. Experimental or simulation-based validation (eg. Molecular Dynamics) could also be used, but would be much more expensive. Additional metrics regarding eg. novelty, synthetic accessibility of generated mixtures would also have to be implemented. Finally, generation would have to handle permutation invariance decoding which is possible. This is a promising line of work that we would want to tackle in the future.
>
> ---
>
> > In lines 44 and 45, physics-based constraints vary across different application domains (such as pharmaceuticals, polymers, etc.). How are these constraints derived? And how is Equation 1 incorporated into the loss function?
>
> We appreciate the reviewer’s comment regarding the use and derivation of physics-based constraints. Physics-based constraints are derived from fundamental governing equations or established empirical laws relating to measured physical properties. For example, in pharmaceuticals, reaction kinetics or degradation rates can often be expressed through transition-state theory or temperature-dependent stability equations. In polymers, viscoelastic and diffusion-driven transport properties follow constitutive relations or diffusion laws based on Fick’s equations solutions.  These domain-specific formulations can be embedded similarly to enhance predictive robustness by leveraging prior scientific knowledge.
>
> In this study, we employ an Arrhenius-based formulation because many thermo-electrical transport properties—such as viscosity (NIST dataset) and ionic conductivity (IlThermo dataset) follow thermally activated behavior with an exponential temperature dependence. Regarding the incorporation of Equation 1, we do not use it into the loss. Instead, the prediction head output parameters which are then fed to the equation to obtain a prediction instead of directly predicting a value. The loss for the Deep Learning model remains unchanged.
>
> ---
>
> > Are the 3D conformations of molecules not considered in the benchmark? Is the 3D conformation also important for the prediction of interactions?
>
> The 3D conformations of molecules are not considered in CheMixHub for the following reasons:
>
> * The majority of the datasets covered are derived from experimental measurements which do not yield such information explicitely.
> * Conformer ensemble modeling methods that work well for single molecules introduce unique challenges when extended to mixtures: While some studies [1] have proposed augmenting the training data by including multiple conformers of the same molecule with identical labels, such technique becomes problematic in the context of mixtures because the number of augmented data points grows exponentially with the number of components and their conformers (e.g., for a binary mixture, all conformer combinations across both components would need to be enumerated).
> * The benefits of incorporating 3D information through conformer ensemble modeling are most pronounced for highly flexible molecules, particularly those with a large number of rotatable bonds. In the datasets we consider, the majority of molecules possess fewer than five rotatable bonds, which limits the expected benefit of conformer ensemble approaches in our context.
>
> A more systematic investigation of conformer ensemble modeling for mixtures in the presence of highly flexible molecules would be an interesting topic for future work and could form the basis of a separate study.
>
> [1] Zhu, Y., Hwang, J., Adams, K., Liu, Z., Nan, B., Stenfors, B., ... & Wang, W. (2023). Learning over molecular conformer ensembles: Datasets and benchmarks. arXiv preprint arXiv:2310.00115
>
> ---
>
> We hope that our response and the improvements we have made to this work after the reviewer feedback bring to the reviewer a new perspective on this work and have mitigated their confusion. If so, we would appreciate it if the reviewer could raise their score.

---

> > ### Comment · Reviewer_QjXn · 2025-08-07
> >
> > Thank you for your reply, which has resolved my confusion. Taking into account the comments of other reviewers, I have decided to increase the score.

---

### Author Response · Authors · 2025-08-06
**General Summary of Rebuttal contributions**

We would like to thank again all of the reviewers for their insightful suggestions and their time. In light of their feedback, we made new contributions to the manuscript which we summarize below.

1. **We improved the descriptions of the data** by providing additional statistics on each of the 11 CheMixHub Tasks.
2. **We concluded that molecular graph representation should be sufficient to capture the relevant shape-determining information** This is due to the high rigidity of the molecules (ie. low number of rotatable bonds) in the datasets considered in the current version of CheMixHub. We leave the study of the impact of 3D conformations information on property prediction of mixtures of highly flexible molecules for future work.
3. **We have set up additional experiments to benchmark zero-shot and transfer learning capabilities of models.** Our results show that fine-tuning effectiveness is strongly influenced by the nature of the original training task: 1) Training on harder tasks (eg. $\Delta H_{mix}$ and $\Delta H_{vap}$) seems to help learning easier tasks (eg. density $\rho$) when considering the same set of molecular mixtures, although architectural differences may influence this conclusion. 2) simple "cross-dataset" fine-tuning approaches underperform compared to "in-dataset" finetuning, highlighting open research questions around how to enable effective multi-task learning for chemical mixture datasets. Regarding zero-shot performances, we observe good zero shot capabilities when predicting on other tasks with the same target property (eg. Viscosity $\eta$) that have similar value ranges.

All of these additional contributions will be added to the camera-ready version. The code is open sourced on our GitHub.

We hope that these contributions clear any confusion and bring additional perspective on this work. If the reviewers have additional questions, we are happy to further discuss our findings. Otherwise, if the reviewers are satisfied with our response to their questions we would appreciate it if the reviewers could raise their score.

---

### Decision · Program_Chairs · 2025-09-18

**Decision:**

Accept (poster)

**Comment:**

(a) Summary of Scientific Claims and Findings
This paper introduces CheMixHub, a comprehensive dataset and benchmarking platform for machine learning on chemical mixtures. It consolidates multiple public sources into 11 well-defined regression tasks spanning diverse application domains such as pharmaceuticals, battery electrolytes, and fuel mixtures. The benchmark includes multiple splitting strategies to evaluate generalization, baseline experiments across classical ML models, graph neural networks, and chemical language models, and integration of physics-based constraints. Together, these contributions establish the first standardized benchmark designed specifically for mixture property prediction, addressing a key gap in molecular ML research.

(b) Strengths
The reviewers agreed that the paper makes a strong and timely contribution by standardizing mixture modeling tasks, which are of high industrial and scientific relevance but have been relatively neglected in ML research. Strengths include:

+ The breadth and diversity of tasks, covering a wide range of real-world mixture applications.
+ The design of robust evaluation protocols, with splits that test extrapolation to unseen components, larger mixture complexity, and out-of-distribution conditions.
+ A systematic modeling space, including both classical baselines and modern deep learning methods.
+ Physics-guided modeling via Arrhenius-based formulations, which improves robustness and interpretability.
+ Open-sourced code and data, with careful documentation, ensuring reproducibility and accessibility.

These qualities align very well with the Datasets and Benchmarks track CFP, which encourages responsible dataset development, benchmark design, and resources to enable future research.

(c) Weaknesses
Several limitations were discussed:
- Lack of inverse modeling or generative tasks (e.g., mixture design from target properties).
- Interpretability of ML models remains limited; no detailed analysis of feature contributions was provided.
- No closed-loop validation against experimental or simulation data, leaving open questions of real-world applicability.
- The original submission did not explore transfer learning or zero-shot scenarios, which are important in practical settings.
- The choice to exclude 3D conformational information may limit performance on flexible molecules, though the authors provided justification for its limited relevance in this dataset.

These weaknesses are reasonable but do not undermine the value of the dataset and benchmark itself, which fits clearly within DB track expectations.

(d) Reasons for Acceptance
The paper fills a clear gap by establishing the first large-scale benchmark tailored to chemical mixtures, a domain with significant scientific and industrial importance. Its standardized framework, reproducible baselines, and integration of physical principles provide an important foundation for future research. While there are natural extensions (inverse modeling, interpretability, simulation validation), they are beyond the reasonable scope of an initial benchmark contribution. The overall strengths—including novelty, breadth, and impact—outweigh the limitations, making this a strong fit for acceptance in the DB track.

(e) Discussion and Rebuttal
The rebuttal period was productive and addressed reviewer concerns directly:

Reviewer QjXn raised questions on data statistics, permutation invariance, multiple properties per composition, 3D conformations, and physics-based constraints. The authors provided detailed clarifications, additional dataset statistics, and justification for the exclusion of 3D conformers. They also clarified the role of physical constraints. This resolved the reviewer’s concerns, leading to an increased score.

Reviewer MoLL raised points about extending tasks to inverse modeling, incorporating broader physical priors, improving interpretability, and including external validation. The authors acknowledged these as promising directions for future work and added performance distribution plots and task sources to improve clarity. The reviewer maintained an accept recommendation.

Reviewer uojg suggested exploring 3D conformers. The authors argued that the dataset’s molecules are largely rigid and showed why 2D representations are sufficient for now. The reviewer accepted this explanation and retained a strong positive rating.

Reviewer VtoY questioned transferability across tasks and the inclusion of zero-shot evaluation. In response, the authors added new experiments on transfer learning and zero-shot prediction, showing nuanced insights about when transfer succeeds (within related tasks) or fails (across unrelated datasets). Despite these additions, the reviewer maintained a borderline score, but this was weighed against the strong consensus of the other reviewers.

Overall, the rebuttal strengthened the paper with new analyses and clearer explanations. The majority of reviewers moved toward or remained at acceptance, and the discussion confirmed the contribution’s importance and scope fit.

Final Recommendation: The paper makes a valuable, timely, and well-aligned contribution to the DB track. I recommend acceptance.